# A climate-change attribution retrospective of some impactful weather extremes of 2021

**Davide Faranda**[1,2,3], **Stella Bourdin**[1], **Mireia Ginesta**[1], **Meriem Krouma**[1], **Robin Noyelle**[1], **Flavio Pons**[1], **Pascal Yiou**[1], **and Gabriele Messori**[4,5,6,7]

[1]Laboratoire des Sciences du Climat et de l'Environnement, UMR 8212 CEA-CNRS-UVSQ,
Université Paris-Saclay, IPSL, 91191 Gif-sur-Yvette, France
[2]London Mathematical Laboratory, 8 Margravine Gardens London, London W6 8RH, UK
[3]LMD/IPSL, École Normale Supérieure, PSL Research University, Paris, France
[4]Department of Earth Sciences, Uppsala University, Uppsala, Sweden
[5]Centre of Natural Hazards and Disaster Science (CNDS), Uppsala, Sweden
[6]Department of Meteorology, Stockholm University, Stockholm, Sweden
[7]Bolin Centre for Climate Research, Stockholm, Sweden

**Correspondence:** Davide Faranda (davide.faranda@cea.fr)

**Abstract.** The IPCC AR6 report outlines a general consensus that anthropogenic climate change is modifying the frequency and intensity of extreme events such as cold spells, heat waves, storms or floods. A pertinent question is then whether climate change may have affected the characteristics of a specific extreme event or whether such event would have even been possible in the absence of climate change. Here, we address this question by performing an attribution of some major extreme events that occurred in 2021 over Europe and North America: the Winter Storm Filomena, the French spring cold spell, the Westphalia floods, the Mediterranean summer heat wave, Hurricane Ida, the Po Valley tornado outbreak, Medicane Apollo and the late-autumn Scandinavian cold spell. We focus on the role of the atmospheric circulation associated with the events and its typicality in present (factual world) and past climate conditions (counterfactual world) – defined using the ERA5 dataset 1950 to present. We first identify the most similar sea-level pressure patterns to the extreme events of interest in the factual and counterfactual worlds – so-called analogues. We then compute significant shifts in the spatial characteristics, persistence, predictability, seasonality and other characteristics of these analogues. We also diagnose whether in the present climate the analogues of the studied events lead to warmer/cooler or dryer/wetter conditions than in the past. Finally

we verify whether the El Niño–Southern Oscillation and the Atlantic Multidecadal Oscillation may explain interdecadal changes in the analogues' characteristics. We find that most of the extreme events we investigate are significantly modified in the present climate with respect to the past, because of changes in the location, persistence and/or seasonality of cyclonic/anticyclonic patterns in the sea-level pressure analogues. One of the events, Medicane Apollo, appears to be a black swan of the atmospheric circulation, with poor-quality analogues. Our approach, complementary to the statistical extreme-event attribution methods in the literature, points to the potentially important role of the atmospheric circulation in attribution studies.

## 1 Introduction

One of the main novelties of the latest IPCC AR6 report (IPCC, 2021) with respect to previous IPCC documents is the increased confidence that anthropogenic climate change is critically affecting weather extremes. As stated by the IPCC AR6,

> a warmer climate will intensify very wet and very dry weather and climate events and seasons, with implications for flooding or drought (high confi-

dence), but the location and frequency of these events depend on projected changes in regional atmospheric circulation, including monsoons and mid-latitude storm tracks.

Similarly, the already very clear statements of the previous reports on changes in temperature extremes are confirmed and strengthened:

> In all continental regions [...] and at the continental scale, it is very likely that the intensity and frequency of hot extremes will increase and the intensity and frequency of cold extremes will decrease.

Other studies underline that we are already observing prolonged periods of extremely warm conditions (Horton et al., 2016) with increased droughts leading to forest fires (Flannigan et al., 2000), species extinctions (Román-Palacios and Wiens, 2020) and health issues for vulnerable populations (Mitchell et al., 2016). Recent scientific literature points to the need of understanding the role of dynamical drivers of changes in weather extremes: in winter, increased persistence of cyclonic and anticyclonic structures can lead to extremely wet or dry periods (Berkovic and Raveh-Rubin, 2022) on the eastern Mediterranean. Further changes in persistence of synoptic structures are also expected under continued global warming in the Northern Hemisphere summer (see, e.g., Kornhuber and Tamarin-Brodsky, 2021). Under global warming, Gordon et al. (2005), Bala et al. (2010) and Pendergrass et al. (2017) suggest that, in the shoulder seasons, we observe a large variability of rains associated with both tropical and extratropical storms and convective events, leading to an alteration of the hydrological cycle.

While these assessments are meaningful when considering (relatively) large ensembles of extreme events with similar characteristics, it is also important to evaluate whether the probability of occurrence or physical characteristics of single extreme events have been influenced by anthropogenic climate change. This knowledge builds awareness of the consequences of greenhouse gas emissions in the general public and allows stakeholders to evaluate specific impacts induced by climate change. For these reasons, attributing a single extreme event to climate change has given rise to a wealth of studies – an entire field named attribution (Shepherd, 2016; Knutson et al., 2017; Jézéquel et al., 2018b; Naveau et al., 2020; van Oldenborgh et al., 2021).

Studies in extreme-event attribution are conventionally grounded in extreme value theory (Trenberth et al., 2015), which they use to estimate return times of threshold exceedances of particular observables (e.g., temperatures above or below a target value for a certain number of consecutive days for heat waves or cold spells). The main drawback of such statistical attribution is that it does not take into account the physical processes leading to the extreme events. Climate change is likely associated with complex dynamical changes in the atmosphere (e.g., Kennedy et al., 2016; Sharmila and

Walsh, 2018; Stendel et al., 2021), yet the conventional extreme value approach overlooks these entirely. This brought Shepherd (2014) to argue that the atmospheric circulation is a key element of the uncertainty in attribution studies and in parallel stimulated attempts to incorporate knowledge of the atmospheric circulation into an attribution framework (Shepherd, 2016; Yiou et al., 2017).

Here, we build upon this line of work by performing an attribution of some notable extremes occurring during the 2021 calendar year, based on large-scale atmospheric drivers. In particular, we analyze (i) the Winter Storm Filomena, which caused, in January, heavy snowfall and extremely cold conditions in Spain; (ii) the late winter cold spell that occurred in April 2021 in France with large impacts on vegetation and agriculture; (iii) the July floods in Westphalia, Germany, responsible for the destruction of entire villages, destruction of infrastructure and heavy loss of lives; (iv) the record-breaking temperatures during the August Mediterranean heat wave and the associated wildfires in Greece and Italy; (v) the September Po Valley tornado outbreak; (vi) Hurricane Ida, which caused extensive damage in Louisiana and New York city; (vii) Medicane (Mediterranean hurricane) Apollo, which caused heavy flooding in Sicily in October; and (viii) the November Scandinavian cold spell, which led to record-low temperatures for the season.

In order to attribute these events to climate change, we study the associated atmospheric circulation patterns and we search for pattern recurrences – which we term analogues – in the far (1950–1979) and recent past (1992–2021). Our working hypothesis is that the far past acts as a counterfactual world where the Earth's climate was less heavily influenced by anthropogenic forcing when compared to the recent past (the factual world). Here, we assume that 30 years is a long enough period to average out high-frequency interannual variability of the atmospheric motions. However, it is necessary to control for the effect of lower-frequency and inter-decadal variability, such as that caused, for example, by the Atlantic Multidecadal Oscillation or by low-frequency modulations of the El Niño–Southern Oscillation. If a direct influence of such low-frequency variability is excluded, then changes in analogues between the two periods we consider are attributed to the climate-change signal. We present in Sect. 2 the methodological aspects of this work, introducing in Sect. 3 the relevant assessment metrics. Section 4 contains, for each event, (i) a meteorological description of the event, (ii) a summary of the known impacts of climate change on that event class and (iii) our attribution analyses. Our conclusions are presented in Sect. 5.

## 2 A method for attributing extreme events to climate change which takes into account changes in atmospheric circulation

We study changes in weather patterns associated with extreme events by leveraging the framework of weather analogues (Yiou et al., 2003). We first identify the peak day of each extreme event. We then perform a semi-objective detection of the concurrent large-scale weather pattern using daily average sea-level pressure (slp) from the ERA5 reanalysis database over 1950–2021 (Hersbach et al., 2020). The choice of using slp is motivated by (i) the fact that in the ERA5 reanalysis, this quantity is closely constrained from station observations; (ii) its capability to track and identify extratropical cyclones (Walker et al., 2020); and (iii) the absence of long-term trends in its values but also in the dynamical systems metrics computed on it (Faranda et al., 2019a; see Sect. 3). The semi-objectivity lies in the exact choice of geographical domain over which the pattern is identified. For cyclones, the domain of the analysis can be easily identified as the low-pressure area associated with the storm. For cold spells and heat waves, we follow Stefanon et al. (2012), who have shown that these events have a large-scale dynamical footprint spanning the size of the European continent. For all cases, we have tested that our method is qualitatively insensitive to modest changes in the domain size. We split the ERA5 dataset into two periods: 1950–1979 and 1992–2021. We take the first period to represent a counterfactual world with a weaker anthropogenic influence on climate than the second period, which represents our factual world affected by anthropogenic climate change. To take into account the possible influence of low-frequency modes of natural variability in explaining differences between the two periods, we also consider the possible roles of the El Niño–Southern Oscillation (ENSO) and the Atlantic Multidecadal Oscillation (AMO).

For each period, we scan all the daily average slp geographical maps and select the best 33 analogues, namely the maps minimizing the Euclidean distance with respect to the map of the event itself. The number of 33 corresponds approximately to the smallest 3‰ of Euclidean distances in each subset of our data. We have tested extracting between 25 and 50 analogue maps, without finding any qualitatively large differences in our results. For the factual period, as is common practice in attribution studies, the event itself is removed. Furthermore, we forbid the analogue search in a window of a week centered around the date of the event. We restrict the analogue search to the extended season during which each event occurs (DJFM, MAMJ, JJAS or SOND) or to the seasons relevant for the occurrence of specific extreme events such as hurricanes or tornadoes. This allows us to identify possible seasonality shifts between the counterfactual and factual periods yet prevents conflating the different physical processes which may contribute to a given class of extremes during the warm versus cold seasons. We then compute the average slp map for all analogues in each of the two periods and take the difference between the two averages ($\Delta$slp). To determine significant changes between the analogue maps of the two periods, we adopt a bootstrap procedure which consists of pooling the dates from the two periods together, randomly extracting 33 dates from this pool 1000 times, creating the corresponding difference maps and marking as significant only grid point changes more than 2 standard deviations above or below the mean of the bootstrap sample. We also plot the 2 m temperature (t2m) and daily precipitation rate fields (tp) on the dates of the closest slp analogues, repeating the same bootstrap procedure to identify significant changes. We additionally plot the distributions of several evaluation metrics in the two periods (see Sect. 3). We finally consider the seasonality of the analogues within the relevant season and their association with ENSO and AMO. We conduct the latter analysis using monthly indices computed from the NOAA/ERSSTv5 data and retrieved from KNMI's climate explorer. In particular, the ENSO index is the 3.4 version as defined by Huang et al. (2017), and the AMO index is computed as described in Trenberth and Shea (2006). When the ENSO 3.4 index is positive, it corresponds to El Niño, and when it is negative, it corresponds to La Niña. To assess the significance of changes in factual vs. counterfactual distributions, we conduct in all cases a two-sided Cramér–von Mises test at the 0.05 significance level. If the $p$ value is smaller than 0.05, the null hypothesis ($H = 0$) that the two samples come from the same distribution can be rejected (Anderson, 1962). All relevant figure panels display the $p$ value (pval) and the result of the test $H$ in the title.

## 3 Evaluation metrics

Following Faranda et al. (2020), we define some quantities that support our interpretation of the analogue-based attribution. All of these may then be compared between the counterfactual and factual periods.

- *Analogue quality $Q$*. $Q$ is the average Euclidean distance of a given day from its closest 33 analogues (Faranda et al., 2020). One can then compare $Q$ for the peak day of the extreme event to $Q$ for each analogue of the extreme event. If the value of $Q$ for the extreme event belongs to the same distribution as, or is smaller than, the values of $Q$ for the analogues, then the extreme event has good analogues, and attribution can be performed. If instead the $Q$ for the extreme event is larger than that of the analogue days, then this indicates a highly unusual slp configuration, and the results of the attribution analysis must be interpreted with care. Differences between the counterfactual and factual periods in the value of $Q$ for the peak day of the extreme event indicate whether the atmosphere is visiting states (analogues) that are more or less similar to the map associated with the extreme. Differences in the distribu-

tion of $Q$ for the 33 analogues indicate whether those states are in turn becoming more or less typical of the atmospheric variability. In order to test the homogeneity of the analogues in the two periods, we have computed $Q$ for all days in the factual and counterfactual periods on a wide North Atlantic domain [80° W–50° E and 22.5–70° N] and applied the two-sided Cramér–von Mises test at the 0.05 significance level. The $p$ value found (0.1995) implies that the null hypothesis that the two samples come from the same distribution cannot be rejected, hence supporting our claim of homogeneity.

– *Predictability index D*. Using dynamical systems theory (Freitas et al., 2011, 2016; Lucarini et al., 2016), we can compute the local dimension $D$ of each daily slp map (Faranda et al., 2017, 2019b; see Appendix A). The local dimension is a proxy for the number of degrees of freedom of the field, meaning that the higher $D$, the more unpredictable the temporal evolution of the slp maps will be (Faranda et al., 2017; Messori et al., 2017; Hochman et al., 2019). If the dimension $D$ of the peak day of the extreme event is higher or lower than that of its analogues, then the extreme will be respectively less or more predictable than the closest dynamical situations identified in the data. We compute two values of $D$ for the event, one using the data in the counterfactual period and one using the data in the factual period. As for $Q$, we also compute the distributions of $D$ for all the analogues in each period. This informs on how predictable the extreme event is with respect to its analogues.

– *Persistence index* $\Theta$. Another quantity derived from the dynamical systems theory is the persistence $\Theta$ of a given configuration (Faranda et al., 2017; see Appendix A). The persistence estimates for how many days we are likely to observe a map that is an analogue of the one considered (Moloney et al., 2019). As for $Q$ and $D$, we compute the two values of the persistence for the extreme event in the factual and counterfactual worlds and the corresponding distributions of persistence for the analogues.

– *Seasonality of analogues*. We can count the number of analogues in each month to detect whether there has been a shift of the circulation towards earlier or later months of the season. This can have strong thermodynamic implications, for example, if a circulation leading to large positive temperature anomalies in early spring becomes more common later in the season, when average temperatures are much higher.

– *Association with ENSO and AMO*. To account for the effect of natural interdecadal variability, we analyze the distributions of the ENSO and AMO indices corresponding to analogues of each event in the factual

and counterfactual periods. If the null hypothesis that the two distributions do not differ between the two periods is rejected, it is not possible to exclude that thermodynamic or dynamic differences in the analogues are partly due to these modes of natural variability rather than anthropogenic forcing. On the other hand, if it is not possible to reject the null hypothesis of equal distributions, observed changes in analogues are attributed to human activity. It is worth noting that such null hypothesis of no influence of natural variability is coherent with the view of Trenberth (2011), who argued that

> Past attribution studies of climate change have assumed a null hypothesis of no role of human activities [...] I argue that because global warming is "unequivocal" and "very likely" caused by human activities, the reverse should now be the case. The task, then, could be to prove there is no anthropogenic component to a particular observed change in climate.

See also the discussion in Lloyd and Oreskes (2018) for support of Trenberth's position.

## 4   Results

Our list of 2021 extreme events is not intended to be exhaustive. We cover Europe and North America, and we try to select events that differ in impacts, season and genesis in order to provide a rich overview of the attribution capabilities of our approach but also of its implementation difficulties. We provide in Table 1 the list of the events studied, with the peak day of each extreme used for the analogue search, affected countries, longitude–latitude boxes used for the analogue search, and months used for the analogue search. A graphical representation of the events is provided in Fig. 1.

### 4.1   Winter Storm Filomena

In early January 2021 the weather regime over the Euro-Atlantic sector was characterized by a negative phase of the North Atlantic Oscillation (NAO), with cold air from the Arctic being advected over southern Europe and frontal activity favored over the Azores. Filomena was associated with an extratropical cyclone that moved from the Azores to the Canary Islands and the Iberian Peninsula on 6 and 7 January respectively, resulting in strong precipitation and hurricane-force winds. It triggered historic snowfalls in the inland regions of the peninsula and a 14 d long cold spell. This exceptional event caused four casualties between 9 and 16 January and economic losses of up to EUR 2 billion (Aon, 2021). The cyclone formed on 1 January in the northeastern inland of the United States. On 3 January it entered the North Atlantic, and it began a sharp displacement southeastward forced by a high-pressure system in the central North Atlantic and

**Table 1.** List of the events presented in this study, with the peak day of each extreme used for the analogue search, affected countries, longitude–latitude boxes used for the analogue search, and months used for the analogue search.

| Event | Date (dd-mm-yyyy) | Countries | Analogue box | Analogue months |
|---|---|---|---|---|
| Winter Storm Filomena | 09-01-2021 | Spain | [15° W, 10° E, 30° N, 46° N] | DJFM |
| French spring cold spell | 06-04-2021 | France | [10° W, 30° E, 30° N, 70° N] | MAMJ |
| Westphalia floods | 14-07-2021 | Benelux/Germany | [5° W, 23° E, 41° N, 59° N] | JJAS |
| Mediterranean heat wave | 11-08-2021 | Spain/France/Italy | [10° W, 25° E, 30° N, 45° N] | JJAS |
| Hurricane Ida | 02-09-2021 | USA | [80° W, 55° W, 35° N, 55° N] | ASON |
| Po Valley tornado outbreak | 19-09-2021 | Italy | [10° W, 20° E, 35° N, 50° N] | MJJASO |
| Medicane Apollo | 29-10-2021 | Italy | [10° E, 20° E, 34° N, 40° N] | SOND |
| Scandinavian cold spell | 28-11-2021 | Sweden/Norway | [10° W, 30° E, 35° N, 75° N] | SOND |

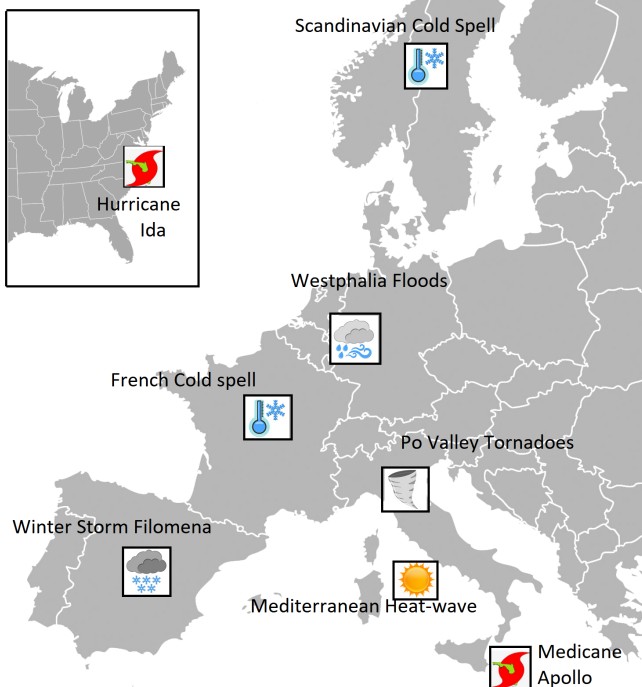

**Figure 1.** Graphical representation of the events studied in this work.

pushed by a polar jet with a strong meridional component. When it arrived west of the Azores on 5 January in somewhat weakened form, it was named Filomena by the Spanish State Meteorological Agency (AEMET), which emitted a severe weather warning for Canary Islands and Spain for the 2 following days. On 6 and 7 January, Filomena strengthened as it moved southeast towards the Canary Islands. The cyclone then traveled northeastward towards the Iberian Peninsula on 7 January, bringing relatively warm, humid air for the winter season. At this time, southern Europe was experiencing cold temperature anomalies because of an anticyclone located west of the UK, resulting in temperature minimums below 0 °C in almost the entire Iberian Peninsula. Hence,

when the storm arrived in the Gulf of Cádiz on 8 January, its warm front blew over the preexisting cold air, allowing precipitation in the form of snow or sleet throughout most of the Iberian Peninsula, except for some parts of southern Spain. The precipitation lasted for 3 d, until Filomena dissipated in the Mediterranean Sea on 11 January. The most affected regions were central and northeastern Spain, which accumulated an average of 30 to 50 cm of snow (AEMET, 2021b). The accumulated snow favored the persistence of low temperatures in the following days, triggering a cold spell that lasted for about 2 weeks, from 5 to 17 January, with a temperature average of 2 °C in the Iberian Peninsula and an anomaly of −3.8° with respect to the 1981–2010 climatology, as recorded by the AEMET (2021b).

### 4.1.1 Extratropical winter storms and climate change

The IPCC report (Lee et al., 2021) highlights that

> the number of extratropical cyclones (ETC) composing the storm tracks is projected to weakly decline in future projections, but by no more than a few percent change

and that

> the reduction is mostly located on the equatorward flank of the storm tracks.

However, it also states that

> substantial uncertainty and thus low confidence remain in projecting regional changes in Northern Hemisphere jet streams and storm tracks, especially for the North Atlantic basin in winter.

Nonetheless, as stated in chap. 11 of the IPCC AR6 (Seneviratne et al., 2021),

> despite small changes in the dynamical intensity of ETCs, there is high confidence that the precipitation associated with ETCs will increase in the future.

In addition, there is

> high confidence that snowfall associated with winter ETCs will decrease in the future, because increases in tropospheric temperatures lead to a lower proportion of precipitation falling as snow.

Besides the IPCC report, numerous studies have addressed the influence of climate change on extratropical cyclones (ETCs) due to their impacts on many regions of the planet (e.g., Zappa et al., 2013; Ulbrich et al., 2009; Priestley and Catto, 2022). Hence, there is a priori mixed evidence for the anthropogenic contribution to dynamical changes in Filomena-like storms.

### 4.1.2 Attribution of Filomena to climate change

We now use the ERA5 data to perform the attribution of the cyclonic circulation associated with Filomena for 9 January 2021 in the past and present climates (Fig. 2). We find a significant increase in the slp up to 3 hPa in the factual period (Fig. 2a–d). Figure 2e–g shows that Filomena was an unusually cold event compared to its analogues even in the counterfactual period. In the factual period analogue temperatures over Iberia are significantly warmer than in the counterfactual period (Fig. 2h), by up to 4 °C. This can likely be related to the long-term surface temperature warming signal in recent years. Precipitation for the factual analogues compared to the counterfactual ones is significantly larger in the center and center-east of the Iberian Peninsula, where Filomena had its highest impact, and in the southeast of the peninsula (Fig. 2l). On the other hand, precipitation is significantly lower in the Gulf of Lion and southwestern Mediterranean Sea.

The analogue quality $Q$ for Filomena is in the upper tail of the distribution, indicating moderate quality, and is poorer in the factual period (Fig. 2m) . There is little change in the event's predictability index $D$ (Fig. 2n) with respect to the atmospheric circulation in the two periods. Still, the analogue distributions in the two periods are statistically different, with the factual period showing a shift towards higher $D$ values. On the contrary, the persistence $\Theta$ of the event with respect to the circulation decreases (Fig. 2o), while no significant change is found in the $\Theta$ of the analogues. Figure 2p shows only modest changes in seasonality, with a slight increase in January and February analogues. Although Filomena occurred during a negative ENSO phase, there is a significant change in the ENSO distribution for the analogues, with the factual period showing more positive values (Fig. 2q). This means that the results may be modulated by ENSO. The distributions of the AMO phases do not evidence any significant influence of this mode on the analogues (Fig. 2r).

Filomena-like storms in the factual period display higher slp yet cause more precipitation in central Spain, the region that suffered the highest impacts from the storm. Even though there are slightly more analogues in the coldest months, that is, January and February, there is a significant increase in the 2 m temperature, making the snow at low altitudes less probable in a warmer climate. Given the reasonable quality of analogues, we can state that the results are in line with the expected climate-change trends discussed in the previous section. However, since there is a shift in the distributions of ENSO conditioned to the analogues, we cannot reject the hypothesis that ENSO variability has some influence on the analogues of Filomena.

### 4.2 French spring cold spell

A frost event took place from 6 to 8 April 2021 in France. It was exceptional, with daily minimum temperatures below −5 °C recorded in several locations. Grapevines and fruit trees were damaged especially in the Loire and Rhône valleys, as frost management strategies (e.g., heating from braziers) could not be implemented in time. The temperatures broke record lows at many French weather stations. This cold event happened 1 week after an episode of high temperatures in March, which was also record-breaking at many locations in France (LaChaineMeteo, 2021) and western Europe. This sequence (or compound event, according to the definition proposed by Zscheischler et al., 2020) led the growing season to start early, with bud burst occurring in March and the new leaves and flowers left exposed to the deep frost episode that followed in early April. The April cold spell was associated with an advection of cold air from the Arctic into France on 5–6 April 2021, facilitated by a deep low pressure based over Scandinavia and anticyclonic conditions overs Iceland. This created the low-temperature anomaly in the subsequent days.

### 4.2.1 Cold spells and climate change

The IPCC AR6 describes as "virtually certain" that there have been warmer and/or rarer cold spells over most land areas since the 1950s, that this trend is due to anthropogenic climate change, and that it is set to continue in the future (IPCC, 2021). As stated in chap. 11 of the IPCC AR6 (Seneviratne et al., 2021),

> a decrease in the number of cold spell days has been observed over nearly all land surface areas (Easterling et al., 2016) and in the northern mid-latitudes in particular (Van Oldenborgh et al., 2019).

While a rapid warming, in general, lowers the probability of cold spell occurrence, projected changes in the temperature distribution imply that regional changes in cold spell frequency and/or intensity may not match changes in the mean temperature (Tamarin-Brodsky et al., 2019). Similarly, Kodra et al. (2011) have shown that long-lasting periods where temperatures drop below an absolute threshold (e.g., frost days) may still be produced locally and occasionally even in future, warmer climates. There has also been a lively debate in

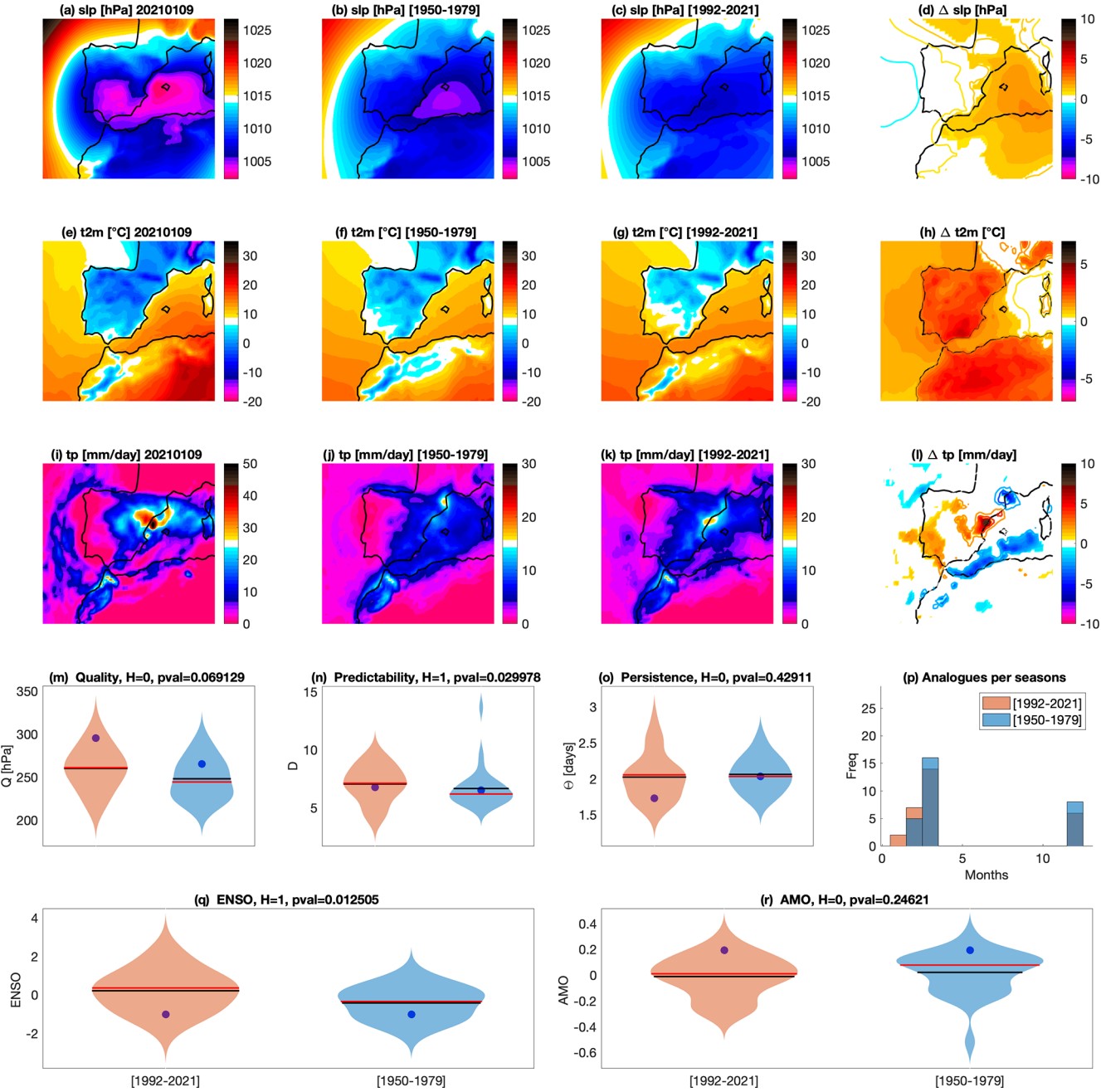

**Figure 2.** Attribution for Storm Filomena on 9 January 2021. Daily mean sea-level pressure slp **(a)**, 2 m temperatures t2m **(e)** and total precipitation tp **(i)** on the day of the event. Average of the 33 sea-level pressure analogues found for the counterfactual [1950–1979] **(b)** and factual [1992–2021] **(c)** periods and corresponding 2 m temperatures **(f, g)** and daily precipitation rate **(j, k)**. Δslp **(d)**, Δt2m **(h)** and Δtp **(l)** between factual and counterfactual periods: colored–filled areas show significant anomalies with respect to the bootstrap procedure. Violin plots for counterfactual (blue) and factual (orange) periods for the analogue quality $Q$ **(m)**, the predictability index $D$ **(n)**, the persistence index $\Theta$ **(o)** and the distribution of analogues in each month **(p)**. Violin plots for counterfactual (blue) and factual (orange) periods for ENSO **(q)** and AMO **(r)**. Values for the peak day of the extreme event are marked by a color-filled circle.

the literature on whether dynamical changes associated with climate change may act to partly counter the thermodynamic changes and favor cold spell occurrence. Faranda (2020) and D'Errico et al. (2022) argued that circulation patterns associated with cold spells over Europe have been increasing in frequency in the present climate and will continue to do so under future climate change. Several authors have also argued for or against a link between Arctic amplification and an increased occurrence of cold spells in some mid-latitude regions (Mori et al., 2014; Cohen et al., 2018; Blackport and Screen, 2020; Ye and Messori, 2020; Jolly et al., 2021).

Cold spells continue to have large detrimental socioeconomic effects, with several high-impact events occurring in recent winters, notably during the 2018–2019 and 2020–2021 winters in North America (Lee and Butler, 2020; BBC, 2022; Lillo et al., 2021; Doss-Gollin et al., 2021; Miller, 2022) and the 2017–2018 winter in Europe (Kautz et al., 2020; LeMonde, 2018). Moreover, even if the absolute severity of cold spells decreases, rapid temperature swings are a hazard in their own right (Kral-O'Brien et al., 2019; Casson et al., 2019).

### 4.2.2 Attribution of the French spring cold spell to climate change

A statistical analysis of the temperatures during the French cold spell of 2021 was proposed by a team of the World Weather Attribution (Vautard et al., 2021). This report concluded that while climate change has raised the absolute temperatures during cold spells, it has also led to an intensification of growing-period frosts due to earlier bud burst. The 2021 cold outbreak occurred right after a specific weather pattern called the "Atlantic Ridge", identified as one of the four main weather regimes in the North Atlantic region (Michelangeli et al., 1995). The goal of this section is to analyze how the features of this weather pattern have evolved with climate change using the ERA5 reanalyses (Fig. 3). This analysis complements the report of Vautard et al. (2021) by examining the atmospheric circulation. We focus on the date of 6 April 2021, the day where the circulation particularly favored the advection of cold air into France. For this day the slp pattern (Fig. 3a) consisted of a ridge of high pressure over the Atlantic and a large cyclonic structure over Scandinavia, with cold air advection from northern latitudes into France. The analogues associated with this circulation in the counterfactual (Fig. 3b) and factual (Fig. 3c) periods exhibit the same zonal pressure gradient, and their difference (Fig. 3d) shows that the gradient is amplified in factual world, leading to stronger cold advection towards France. The t2m for 6 April 2021 (Fig. 3e) shows cold conditions over northern and western Europe, while the analogues are milder (Fig. 3f, g), and $\Delta$t2m is mostly greater than $0\,°C$ everywhere. If we focus over France, we can conclude that this cold spell would have led to temperatures 2–$4\,°C$ colder without anthropogenic forcing. Looking at the

precipitation maps (Fig. 3i, j, k) and the $\Delta$tp (Fig. 3l) we see that the cold spell atmospheric pattern corresponds to dry conditions over France. There is no change in precipitation patterns over France between the factual and counterfactual conditions (Fig. 3l). However, the reinforcement of the zonal pressure gradient in the factual period leads to an increase in the precipitation over continental Europe and a decrease on the Mediterranean Sea. The values of $Q$ (Fig. 3m) suggest that the pattern under examination is rare compared to its analogues. The distribution of the predictability index $D$ (Fig. 3n) shifts towards lower values in the factual period, although there are no significant changes relative to the counterfactual distribution. Similarly, there are not significant shifts in the distribution of the persistence $\Theta$ (Fig. 3o). Nonetheless, the extreme itself becomes markedly more persistent relative to the atmospheric circulation in the factual period. The monthly distribution of the analogues (Fig. 3p) suggests that there is a shift of this circulation pattern towards April and June months and that its occurrence in March is decreasing in recent times.

Figure 3q suggests a significant change in the ENSO phases associated with the analogues in the two periods, while no significant role of the AMO is detected with this analysis (Fig. 3r). Therefore the attribution of this event to climate change comes with the caveat of a potential role of ENSO on the associated pattern of atmospheric circulation.

To conclude, our analysis suggests, in line with the literature on cold spells and climate change cited in Sect. 4.2.1, that the French spring cold spell event is becoming rare in the month of March in the current climate and that it would have led to cooler temperatures in a world without climate change.

### 4.3 Westphalia floods

On 11 July 2021 the synoptic situation over western Europe was characterized by a ridge situated west of Ireland. As this low-pressure system – named "Bernd" by the German Meteorological Service (DWD; see Junghänel et al., 2021) – gradually moved eastward, it was isolated from the usually westerly large-scale flow by a strong anticyclonic system that built up over the eastern part of the Atlantic and deviated the jet stream north of Scotland. By 13 July, Bernd was completely cut from the main flow and remained stationary over western and central Europe until 16 July, before being gradually pushed east. Hot and moist surface air from northern Europe and the Mediterranean was advected by the cyclonic movement around the cutoff, which led from 12 to 15 July to recurrent and persistent heavy rains first over mountain ranges due to orographic and dynamic uplift and then over the entire region of Belgium, Luxembourg, western Germany and eastern France. The maximum precipitations over the region were centered on the west of Belgium with some locations receiving more than $250\,mm$ of rain in $48\,h$ (e.g., in Jalhay, Belgium, according to what reported

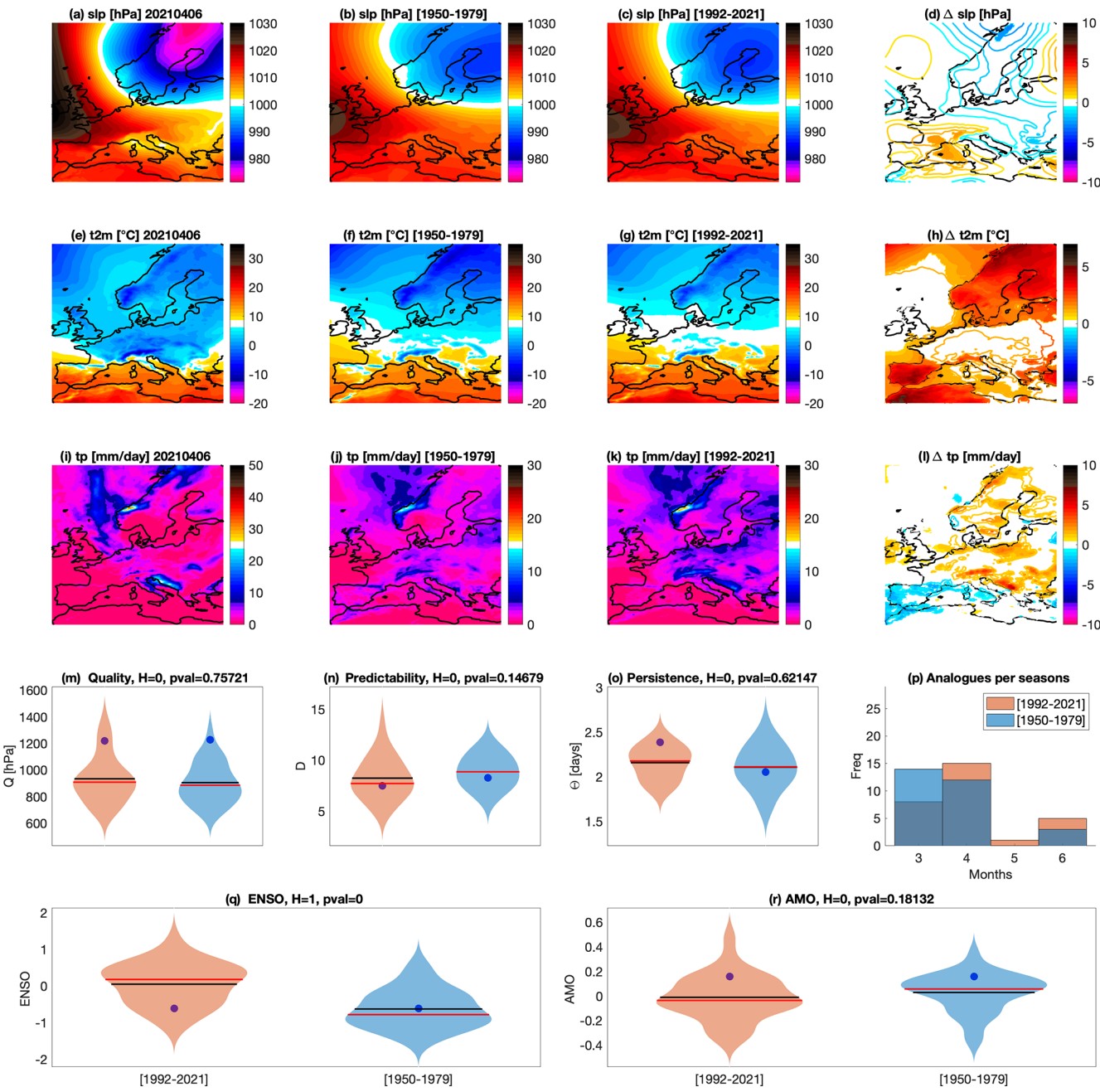

**Figure 3.** Attribution for the French cold spell on 6 April 2021. Daily mean sea-level pressure slp (**a**), 2 m temperatures t2m (**e**) and total precipitation tp (**i**) on the day of the event. Average of the 33 sea-level pressure analogues found for the counterfactual [1950–1979] (**b**) and factual [1992–2021] (**c**) periods and corresponding 2 m temperatures (**f, g**) and daily precipitation rate (**j, k**). Δslp (**d**), Δt2m (**h**) and Δtp (**l**) between factual and counterfactual periods: colored–filled areas show significant anomalies with respect to the bootstrap procedure. Violin plots for counterfactual (blue) and factual (orange) periods for the analogue quality $Q$ (**m**), the predictability index $D$ (**n**), the persistence index $\Theta$ (**o**) and the distribution of analogues in each month (**p**). Violin plots for counterfactual (blue) and factual (orange) periods for ENSO (**q**) and AMO (**r**) indices. Values for the peak day of the extreme event are marked by a blue dot. Horizontal bars in panels (**m**)–(**r**) correspond to the mean (black) and median (red) of the distributions.

by Kreienkamp et al., 2021). The soils, already humid due to recurring precipitation events during the preceding 3 weeks, were incapable of absorbing more water, which led to runoff and overflow of small watercourses and flash floods. Afterwards, larger rivers such as the Ruhr and the Meuse also overflowed, causing massive casualties mainly in Germany (196 people, according to DieWelt, 2021) and Belgium (42 casualties, according to Het Laatste Nieuws, 2021). In addition to the terrible fatalities, the floods severely damaged goods and infrastructure, with a total cost estimated around EUR 10 billion (Business Insurance, 2022) for Belgium. It was afterwards found using hydrological data that the flood in the regions affected was significantly higher than any flood since the beginning of the systematic records (Kreienkamp et al., 2021).

### 4.3.1 Floods and climate change

Rapidly after the event, the potential link between the event and climate change was highlighted by activists and journalists. Indeed, as the atmosphere warms up, it can contain more water – 7 % K$^{-1}$ of warming according to the Clausius–Clapeyron relationship – therefore allowing more intense extreme precipitation events. Several studies (Madsen et al., 2014; Kundzewicz et al., 2018, 2019) investigated the link between climate variability, extreme precipitation, and hydrological floods globally and in Europe. As stated in the last IPCC report (IPCC, 2021), there is high confidence that

> a warmer climate will intensify very wet and very dry weather and climate events and seasons, but the location and frequency of these events depend on projected changes in regional atmospheric circulation.

Especially for Europe, there is medium confidence that at 1.5 °C of warming,

> heavy precipitation and associated flooding are projected to intensify and be more frequent.

This result highly depends on the type of water basins, especially if the peak flow is snowmelt-dominated. More generally, heavy precipitations are strongly entangled with natural variability of the climate system. Ultimately, although flooding usually depends strongly on the local characteristics of the hydrological system – especially artificialization of soils and containment of rivers – more intense flooding can be linked to climate change via the increased intensity of heavy rains.

### 4.3.2 Attribution of Westphalia floods to climate change

An attribution study of the Westphalia floods has already been published by the World Weather Attribution network, which investigated the influence of climate change on heavy precipitations over a broad region of western Europe (Kreienkamp et al., 2021). The authors of the study concluded that a climate warming of 1.2 °C (current climate) led to an increase in the likelihood of such an event by a factor between 1.2 and 9 with respect to the pre-industrial period. Here, we condition the attribution results on the atmospheric dynamics leading to the occurrence of similar events. Results of our attribution analysis are displayed in Fig. 4. We found no significant decrease in the slp of the cutoff low over Germany between the factual and counterfactual periods (Fig. 4a–d) and only moderate increases in t2m over the regions of interest (Fig. 4e–h). We found also a large and significant increase in precipitation (up to 5 mm d$^{-1}$) over southwest Germany, eastern France and the western Alps (Fig. 4i–l). This increase is consistent with the increasing amount of water vapor that a warmer atmosphere can carry.

Overall, the analogue quality (Fig. 4m) is good in both periods. It allows us to emphasize that, even if intense precipitation events due to cutoff lows over western Europe in summer are not unusual, this event was particularly intense and climate change likely made it more intense via an increased quantity of water vapor in the atmosphere. No significant changes are observed in the distributions of predictability $D$ (Fig. 4n) and persistence $\Theta$ (Fig. 4o) or in the predictability or persistence of the event itself relative to the circulation in the two periods. In the factual period, events tend to happen slightly more frequently in the month of July (Fig. 4p), a favorable month for the development of large convective systems in the area, but overall changes in seasonality are small. When investigating the link between the event and low-frequency variability of the climate system, Fig. 4q shows no significant difference in the ENSO distributions during analogue occurrences between the factual and counterfactual periods, even though the distribution in the counterfactual period is broader. Figure 4r, however, displays a significant change between the AMO distributions, with the analogues in the factual period being found in warmer phases of the AMO than in the counterfactual period. This suggests that attributing this event to climate change requires disentangling the possible role of AMO versus global warming.

In summary, the Westphalia floods occurred after an intense rain event caused by a cutoff lows stagnating over the region of Belgium, Luxembourg, western Germany and eastern France. They caused massive casualties and severely damaged property and infrastructure. Our analysis is coherent with the existing literature which shows that a warmer atmosphere leads to an intensification of extreme rain events which in turn can exacerbate the intensity of floods. It should nonetheless be emphasized that attributing this event requires taking into account the role of low-frequency climate variability in the results.

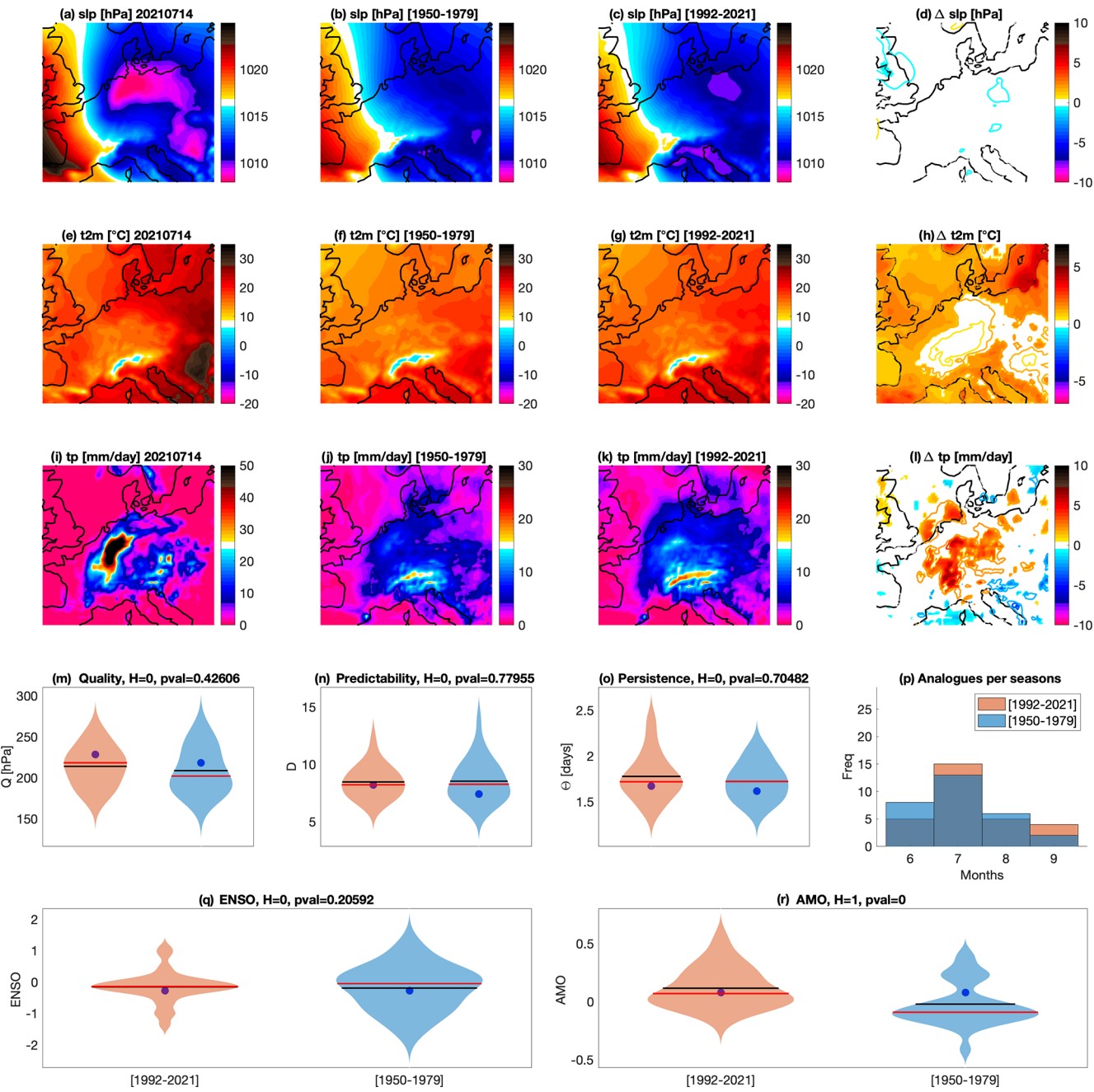

**Figure 4.** Attribution for the Westphalia floods on 14 July 2021. Daily mean sea-level pressure slp **(a)**, 2 m temperatures t2m **(e)** and total precipitation tp **(i)** on the day of the event. Average of the 33 sea-level pressure analogues found for the counterfactual [1950–1979] **(b)** and factual [1992–2021] **(c)** periods and corresponding 2 m temperatures **(f, g)** and daily precipitation rate **(j, k)**. Δslp **(d)**, Δt2m **(h)** and Δtp **(l)** between factual and counterfactual periods: colored–filled areas show significant anomalies with respect to the bootstrap procedure. Violin plots for counterfactual (blue) and factual (orange) periods for the analogue quality $Q$ **(m)**, the predictability index $D$ **(n)**, the persistence index $\Theta$ **(o)** and the distribution of analogues in each month **(p)**. Violin plots for counterfactual (blue) and factual (orange) periods for ENSO **(q)** and AMO **(r)**. Values for the peak day of the extreme event are marked by a blue dot. Horizontal bars in panels **(m)**–**(r)** correspond to the mean (black) and median (red) of the distributions.

## 4.4 Mediterranean heat wave

During the month of August, an area of high pressure in the upper troposphere affected a large part of the Mediterranean basin. The upper-lever high-pressure system caused atmospheric subsidence which simultaneously compressed the air and warmed it, a phenomenon known as "heat dome". This atmospheric configuration induced a severe heat wave over the Mediterranean region from 10 to 15 August: southern Italy, France, Spain and north Africa were most affected, with extensive wildfires and high temperatures. On 11 August, record-breaking temperatures were recorded at several locations in Italy. The town of Santa Maria Capua Vetere in Campania reached 42.2 °C, 44.5 °C was recorded at Bova in Calabria and 43.6 °C was recorded at Ballao in Sardinia (Mazzoleni, 2021). The highest temperature was recorded in eastern Sicily with a peak of 48.8 °C recorded in Floridia in the province of Syracuse (SIAS, 2021). This is the current European temperature record. From 12 August, the heat dome moved towards Spain. There, the heat peak was reached on 14 August, establishing a new national temperature record of 47.4 °C in Montoro, Andalusia (AEMET, 2021a). The heat wave also reached southeastern France, where 40.9 °C was recorded in Varages in the Var, and 41.2 °C was recorded in Trets, Bouches-du-Rhône. Some records were broken also in Tunisia, with 47 °C in Tunis and 50.3 °C in Kairouan (WMO, 2021). The heat wave additionally triggered extensive wildfires in Italy, Spain, France and Greece. During the night of 11 to 12 August, more than 500 fires were recorded in Italy, causing four casualties (CEMS, 2021c). Spain faced fires in the area of Navalacruz and Riofrío. A fire of 90 km of perimeter devastated 12 000 ha of vegetation and led to the evacuation of 1000 inhabitants (CEMS, 2021a). Similarly in the Var (France) wildfires burned 6300 ha and resulted in the evacuation of 7000 people and the death of 2 people (CEMS, 2021b).

### 4.4.1 Mediterranean heat waves and climate change

The IPCC AR6 (Ali et al., 2022) clearly highlights the major changes in heat wave characteristics in the Mediterranean region brought about by climate change. The report states that

> Surface temperature in the Mediterranean region is now 1.5 °C above the pre-industrial level, with a corresponding increase in high-temperature extreme events (high confidence)

and that

> A growing number of observed impacts across the entire basin are now being attributed to climate change, along with major roles of other forcing of environmental change (high confidence). These impacts include multiple consequences of longer and/or more intensive heat waves.

Finally, the report states that

> During the 21st century, climate change is projected to intensify throughout the [Mediterranean] region. Air and sea temperature and their extremes (notably heat waves) are likely to continue to increase more than the global average (high confidence).

Several studies in the literature have investigated the changes related to climatic factors in the Mediterranean, coming to similar conclusions concerning the generalized increase in heat wave frequency and intensity expected in the region (e.g., Guerreiro et al., 2018; Molina et al., 2020), and also highlighting that this may be accompanied by a drying trend (Spinoni et al., 2020; Grillakis, 2019).

### 4.4.2 Attribution of the Mediterranean heat wave to climate change

We use ERA5 to perform the attribution of the anticyclonic circulation associated with the Mediterranean heat wave in past and present climates. We note that we will select the analogues independently of the extratropical or tropical nature of the depression that produced them. Figure 5 shows the results for the 11 August 2021, when the heat wave peaked over southern Italy. We do not detect a significant change in the slp for the factual period compared to the counterfactual period (Fig. 5a–d). However, we do observe a significant warming in t2m in the factual analogues compared to the counterfactual ones (Fig. 5h), with positive $\Delta$t2m anomalies of 2–3 °C over much of the land areas in the western Mediterranean basin. Nonetheless, the factual analogues are still cooler than the observed extremely warm conditions on 11 August 2021 (Fig. 5e, g). The warming in the factual period is associated with a significant decrease in tp in southern continental Europe and over Sicily, which could be explained by the high temperatures and stability which suppress convection (Fig. 5i–l). The $Q$ values (Fig. 5m) suggest a reasonably good analogue quality in both periods. Again in both periods, the extreme-event predictability index $D$ is close to the maximum of the analogue distributions (Fig. 5n), despite the fact that the two distributions are significantly different. This means that the slp pattern for the observed heat wave was unpredictable relative to its analogues. Moreover, the event's $D$ is higher when calculated on the factual period data than on the counterfactual period data. Persistence $\Theta$ shows no significant changes in the analogues' distribution or large changes in the event's $\Theta$ as computed from the data in the two periods (Fig. 5o). Only minor changes in seasonality are observed (Fig. 5p). We finally looked at the possible influence of the low-frequency variability (ENSO and AMO) on the analogues (Fig. 5q, r). We cannot dismiss the impact of ENSO and AMO variability as the distributions of both indices conditioned on the analogues change significantly between the factual and counterfactual periods. Specifically, the

ENSO distribution shifts from weakly negative to neutral values, while the AMO distribution shifts from weakly negative to positive values.

In summary, our analysis is in line with the existing literature cited in Sect. 4.4.1, as it shows the predominance of the thermodynamic effects of climate change on the heat wave, with a clear warming signal in the analogues, which is higher than that of the global average. This signal is associated with dryer conditions over land. We nonetheless reiterate the possible influence of low-frequency climate variability on our results.

## 4.5 Hurricane Ida

Hurricane Ida was a tropical and post-tropical cyclone that occurred in the North Atlantic basin (Caribbean Sea and mainland USA) in August 2021. Besides being the most intense tropical cyclone (TC) to make landfall in the USA in that season, it had a very damaging post-tropical stage. Hurricane Ida (track shown in Fig. 6) was first detected as a tropical wave on 23 August. It was named as a tropical storm on 26 August, and it became a Category 1 hurricane on the day it made a first landfall over Cuba on 27 August. This landfall did not weaken it, and it underwent rapid intensification as it approached Louisiana's coast, where it made landfall again as a Category 4 hurricane (NHC/NOAA, 2021). At its peak intensity, 1 min sustained winds reached $240\,\mathrm{km\,h^{-1}}$ and the minimum central pressure was 929 hPa. Notably, it did not rapidly weaken because of the "brown ocean effect", where flat and moist land conditions allow a TC to retain its intensity for a longer period of time. Ida finally dropped below hurricane strength on 30 August.

While it was still a tropical wave, Ida triggered floods in Venezuela with 20 casualties. In Cuba, the material damage was important, but no casualties were reported. In Louisiana and Mississippi there were a total of 38 deaths, among which 23 were indirect, mostly from carbon monoxide poisoning (Hanchey et al., 2021). A large power outage left more than 1 million experiencing a blackout. Heavy infrastructural damage is estimated around USD 15 billion (NCDC/NOAA, 2021). These figures can be compared to Katrina's – the costliest hurricane to date, which made landfall on the same date and the same place 16 years before – 1838 deaths and USD 125 billion in damages (NHC/NOAA, 2018).

While Ida was weakening into an extratropical low, it combined with a frontal zone, regaining tropical-storm force winds and unleashing large amounts of rainfall over the northeastern USA. The casualties in this region were greater than those for Ida's tropical stage, with 42 deaths mostly due to flash floods. Finally, Ida ended its course over eastern Canada, dissipating in the Gulf of St. Lawrence.

### 4.5.1 Hurricanes and climate change

Of all extreme events, tropical cyclones (TC) are among those for which the impacts of climate change are the most uncertain. The reason for this is threefold: (i) the lack of a satisfying theory for cyclogenesis, (ii) the short span of reliable observations, and (iii) the difficulty to simulate TCs in state-of-the-art global models, because of their too coarse resolution. Despite the relatively short span of available observations, some conclusions can still be drawn from the past record (Knutson et al., 2019).

Notably, the IPCC's AR6 report (IPCC, 2021) states that

> it is very likely that heavy precipitation events will intensify and become more frequent in most regions with additional global warming. At the global scale, extreme daily precipitation events are projected to intensify by about 7 % for each 1 °C of global warming (high confidence). The proportion of intense tropical cyclones (categories 4–5) and peak wind speeds of the most intense tropical cyclones are projected to increase at the global scale with increasing global warming (high confidence). (SPM, B2.4)

Modeling studies using different methodologies (large-scale indicators vs. direct TC tracking) disagree on the sign of future global TC frequency trends. There is nonetheless some confidence in trends of TC-related risks. Knutson et al. (2020) highlight these in order of decreasing certainty: (1) because of sea-level rise, storms surges will become more important; (2) TC precipitation rates will increase; (3) the proportion of intense TCs among all TCs will continue to rise, and the maximum surface wind speed will increase of about 5 %.

There is also growing concern about the increase in windstorm risks associated with post-tropical cyclones (Haarsma, 2021). Indeed, studies in reanalyses showed that despite representing a small number of extratropical storms, post-tropical cyclones are among the most intense ones to reach North America and Europe (Baker et al., 2021; Sainsbury et al., 2020). A global climate-change projection shows that more tropical cyclones are likely to undergo post-tropical transition in the future, especially in the North Atlantic basin (Michaelis and Lackmann, 2019).

### 4.5.2 Attribution of Hurricane Ida to climate change

We now focus on the day Ida produced heavy precipitation in New York City, namely 2 September 2021, and apply the analogue methodology to perform an attribution. We note that we select analogues independently of the extratropical or tropical nature of the depression that has produced them. Figure 7a shows the daily slp associated with Ida on the chosen date, and Fig. 7b and c show the analogue average for the counterfactual and the factual periods. We find a signifi-

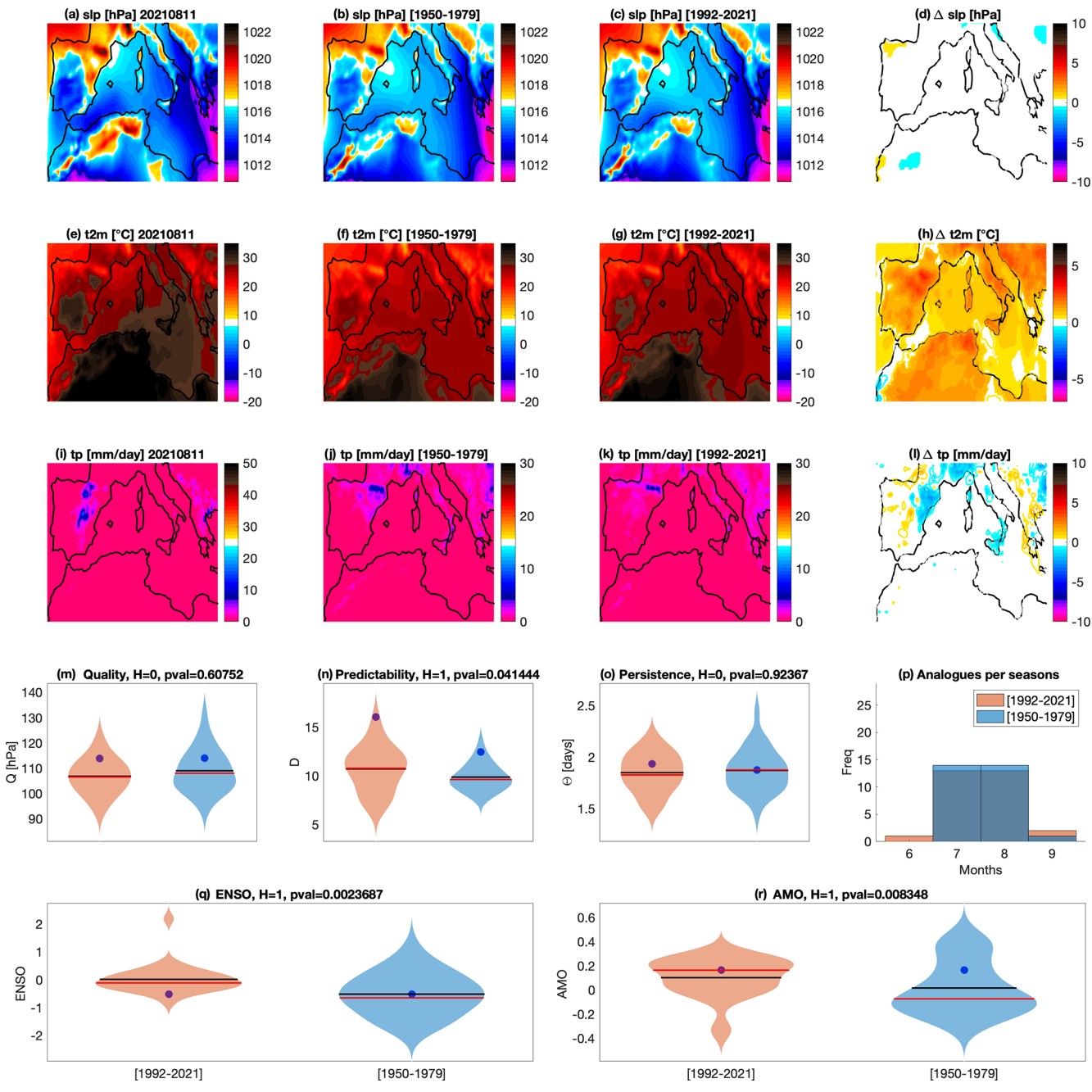

**Figure 5.** Attribution for the Mediterranean heat peak on 11 August 2021. Daily mean sea-level pressure slp **(a)**, 2 m temperatures t2m **(e)** and total precipitation tp **(i)** on the day of the event. Average of the 33 sea-level pressure analogues found for the counterfactual [1950–1979] **(b)** and factual [1992–2021] **(c)** periods and corresponding 2 m temperatures **(f, g)** and daily precipitation rate **(j, k)**. Δslp **(d)**, Δt2m **(h)** and Δtp **(l)** between factual and counterfactual periods: colored–filled areas show significant anomalies with respect to the bootstrap procedure. Violin plots for counterfactual (blue) and factual (orange) periods for the analogue quality $Q$ **(m)**, the predictability index $D$ **(n)**, the persistence index $\Theta$ **(o)** and the distribution of analogues in each month **(p)**. Violin plots for counterfactual (blue) and factual (orange) periods for ENSO **(q)** and AMO **(r)**. Values for the peak day of the extreme event are marked by a blue dot. Horizontal bars in panels **(m)**–**(r)** correspond to the mean (black) and median (red) of the distributions.

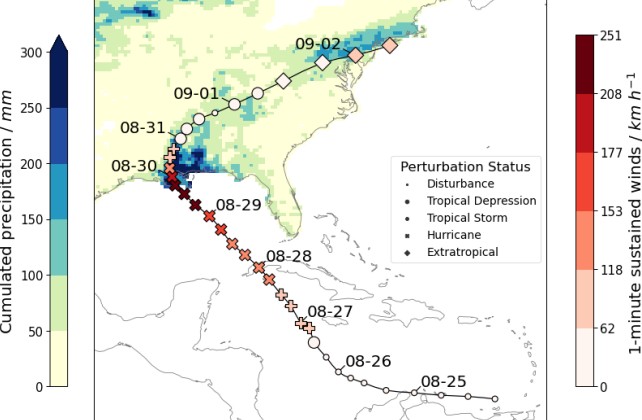

**Figure 6.** Track and associated precipitation for Hurricane Ida. The 6-hourly track positions from the IBTrACS (Knapp et al., 2010; Knapp et al., 2018) database are provided with their wind speed and status from the NHC report. Cumulated daily precipitation between 28 August and 3 September 2021 from the NCEP/CPC (2022) US Unified Precipitation is displayed. White indicates no data.

cant weakening of the slp depression (i.e., an increase in the minimum slp) for the factual with respect to the counterfactual period (Fig. 7d). Furthermore, we observe that temperatures (Fig. 7e–g) are significantly warmer (Fig. 7h) in the factual period. The signal of changes in analogues' precipitation between the two periods is mixed, and both sets of analogues additionally display relatively different precipitation patterns from that observed for Ida (Fig. 7i–l). We have confidence in these results because the quality of the analogues $Q$ for the event is well within (albeit in the upper tails of) the distributions of $Q$ for its analogues in both factual and counterfactual periods (Fig. 7m). The distribution of analogue quality changes significantly between the two periods, and we observe that the distribution is narrower and shifted towards lower values in the factual period, meaning that the event is becoming more typical (Fig. 7m). There is also a significant change in predictability of the analogue distribution, with a shift towards lower $D$ (and hence higher predictability, Fig. 7n), but not in persistence (Fig. 7o). We see an increase in analogues in the months of August/September in the factual period (Fig. 7p): these months are in the tropical cyclone season in the North Atlantic, and therefore it is likely that more events in the factual period correspond to post-tropical cyclones. This is in line with the significant change in the AMO distribution between the two sets of analogues, with a shift towards more positive (warmer) values in the factual period (Fig. 7r). Indeed, a warmer phase of the AMO favors cyclonic activity and hence post-cyclonic activity. There is no significant change in the distribution of ENSO between the two sets of analogues (Fig. 7q).

Ida was already a rare extreme event as a Category 4 hurricane, but it will leave a mark especially because of its impactful post-tropical stage. As discussed in Sect. 4.5.1, very intense hurricanes are likely to become more frequent with climate change, and they will be more likely to undergo post-tropical transition. What is particular for Ida, however, is that this transition occurred inland. What allowed the storm to remain intense in between a very strong tropical cyclone stage and the encounter with an extratropical perturbation could be the wet and warm conditions allowing for the brown ocean effect. However, we are aware of no formal study of such inland post-tropical cyclones in the literature. An important caveat of our analysis is that it does not take into account the post-tropical or extratropical nature of the analogue storms but only their slp footprints, so that it is hard to disentangle changes in the type of events that would occur in the area and the impact of climate change on each type of event. Our results nonetheless highlight a potential increase in autumn storm risk over northeastern North America and a possible AMO-driven modulation in the observed signal.

## 4.6 Po Valley tornado outbreak

On 19 September 2021, an outbreak of seven tornadoes affected the central Po Valley, in northern Italy. In particular, six of these formed in Lombardy and one, the most intense and damaging, hit a small airport near Carpi, Emilia-Romagna. Both mesocyclonic and non-mesocyclonic vortices were observed during the event, one the most impressive tornado outbreaks on record for the region. While tornadoes and waterspouts do occur regularly in Italy, they are on average much less frequent and less intense than in areas such as the midwestern and southeastern USA. However, the structure and location of the Po Valley can lead to the insurgence of environmental conditions conducive for occasionally intense phenomena, including tornadoes reaching EF4+ intensity on the Enhanced Fujita scale (Doswell et al., 2009). During the summer, the Po Valley can persistently host hot and humid air. The presence of the Adriatic Sea to the southeast provides an additional source of moisture, which can be advected to the region by the low-level jet preceding low-pressure systems approaching from the northwest. Moreover, the presence of the Apennines mountain range can encourage the formation of dry lines in the event of southwesterly flow due to foehn effect, contributing to supercell development (Alberoni et al., 1996). On 19 September, a high-pressure system extended from the central Mediterranean Sea to Scandinavia, while a high-level low pressure approached the Po Valley from France, connected to a trough located over northwestern Europe. During the afternoon, the region was affected by a dynamic and thermodynamic setup favorable to tornado development: a hot and humid low-level jet from the east, a strong wind shear with winds from the southwest at 500 hPa, a jet stream from the west at 200 hPa, and an approaching upper-level low characterized by relatively cold air and by the entrainment of stratospheric dry air. This led to the formation of strong thunderstorms associated with six tornadoes over Lombardy, roughly arranged

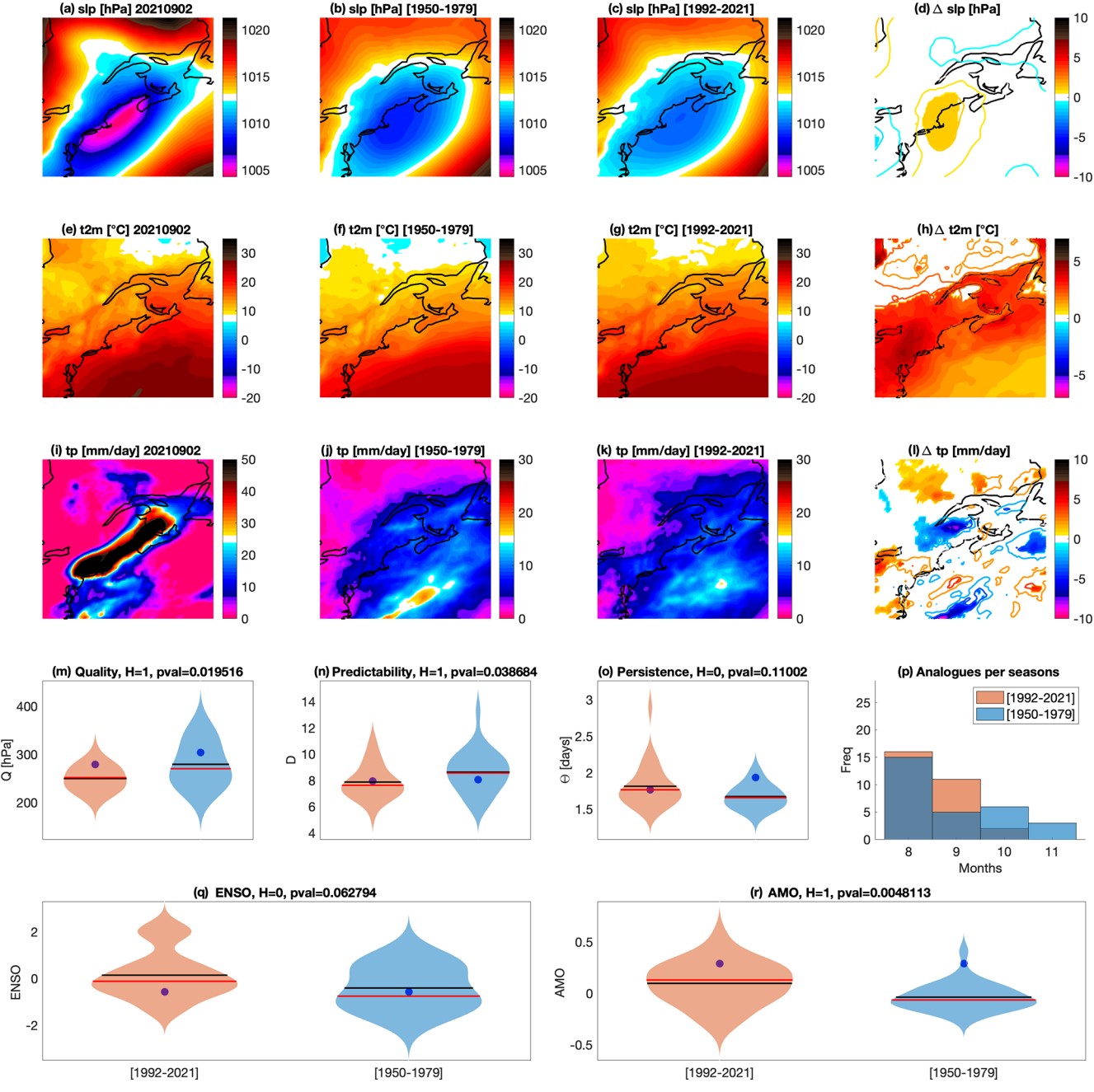

**Figure 7.** Attribution for the Hurricane Ida passage over the New York City area on 2 September 2021. Daily mean sea-level pressure slp **(a)**, 2 m temperatures t2m **(e)** and total precipitation tp **(i)** on the day of the event. Average of the 33 sea-level pressure analogues found for the counterfactual [1950–1979] **(b)** and factual [1992–2021] **(c)** periods and corresponding 2 m temperatures **(f, g)** and daily precipitation rate **(j, k)**. △slp **(d)**, △t2m **(h)** and △tp **(l)** between factual and counterfactual periods: colored–filled areas show significant anomalies with respect to the bootstrap procedure. Violin plots for counterfactual (blue) and factual (orange) periods for the analogue quality $Q$ **(m)**, the predictability index $D$ **(n)**, the persistence index $\Theta$ **(o)** and the distribution of analogues in each month **(p)**. Violin plots for counterfactual (blue) and factual (orange) periods for ENSO **(q)** and AMO **(r)**. Values for the peak day of the extreme event are marked by a blue dot. Horizontal bars in panels **(m)**–**(r)** correspond to the mean (black) and median (red) of the distributions.

along a line between the cities of Milan and Brescia. Around 17:00 CEST (UTC+2), an isolated thunderstorm formed to the southeast of this area, closer to the Apennines range, and assumed markedly supercellular features, with a hook-echo reflectivity signature, a Doppler velocity couplet and a deviation to the right with respect to the mid-level flow: all clear signs of a strong rotating updraft. This supercell produced a well-documented tornado which hit a local airport, resulting in possible EF3 damage (Poli and Stanzani, 2022).

### 4.6.1 Tornadoes and climate change

The IPCC AR6, chap. 11 (Seneviratne et al., 2021), states that past trends in tornado occurrence are not robust due to short observation time series and that

> There is medium confidence that the mean annual number of tornadoes in the USA has remained relatively constant, but their variability of occurrence has increased since the 1970s, particularly over the 2000s, with a decrease in the number of days per year, and an increase in the number of tornadoes on these days (high confidence). Detected tornadoes have also increased in Europe, but the trend depends on the density of observations.

Moreover, even though high confidence is given to an increase in CAPE over the tropics and subtropics, over the USA, the increase in CAPE could be associated with a decrease in the vertical wind shear. This according to the IPCC suggests

> favourable conditions for an increase in severe convective storms in the future, but the interpretation of how tornadoes or hail will change is an open question because of the strong dependence on shear.

Finally, the IPCC report (Seneviratne et al., 2021) concludes that it is

> extremely difficult to detect and attribute changes in severe convective storms.

Most studies are focused on the USA, pointing to an increased variability, efficiency and possibly intensity of tornado outbreaks in the last decades (Brooks et al., 2014; Elsner et al., 2015, 2019). However, tornadoes in Europe remain an underestimated threat (Antonescu et al., 2017), even though they can affect very densely populated areas, as in the case described in this article.

### 4.6.2 Attribution of the Po Valley tornado outbreaks to climate change

Figure 8 shows the results for the attribution of the synoptic configuration associated with the Po Valley tornado outbreak episode. We do not observe significant differences in the pressure field over the Po Valley and only a marginally weaker low-pressure area in the Genoa Gulf (Fig. 8a–d) for the factual with respect to the counterfactual analogues. Instead, we observe that temperatures are significantly warmer (Fig. 8h) in the recent period, especially over land, including the Po Valley, and the Adriatic sea. This provides an increased amount of convective potential energy, through the transport of hot and humid air within the low-level jet. The factual period atmospheric configuration is further associated with higher precipitation over the Alps and central Europe and slightly lower precipitation over the Italian Peninsula (Fig. 8i–l), which is coherent with a more intense transport of warm and humid air from the southeast.

The analogue quality shows that this circulation pattern is relatively common compared to the rest of the analogues. We do not detect visible changes in the predictability $D$ (Fig. 8n) and persistence $\Theta$ (Fig. 8o) of the analogues between the two periods. However, the predictability of the event itself is lower (higher $D$) when computed using data from the factual period. The seasonal occurrence of analogues (Fig. 8p) is quite consistent with the months of occurrence of tornadoes in northern Italy, with a maximum during summer; however, we do observe a general shift towards analogues occurring earlier in the season during the factual period, with the largest increase in July, when land-surface temperatures reach the annual maximum and the probability of low-pressure areas entering the Mediterranean basin is higher than in May or June, offering more energy and occasions for convective instability. Finally, changes in the distributions of ENSO (Fig. 8q) and AMO between the two periods (Fig. 8r) are at the very margin of statistical significance, suggesting that no strong conclusion can be drawn on the influence or lack thereof of these modes of decadal and inter-decadal variability.

Our analysis of the Po Valley tornado outbreak shows a clear increase in temperature of the analogues of this event in the factual period. This is compatible with the occurrence of more favorable environments for tornadoes due to climate change as mentioned in Sect. 4.6.1. However, the small spatiotemporal scale of the phenomenon requires caution in the interpretation of the attribution results.

## 4.7 Medicane Apollo

When the relatively cold atmospheric air coming from polar latitudes meets the warm surface of the Mediterranean Sea, extratropical cyclones change their characteristics into near-tropical depressions. These hybrids – termed "medicanes" (portmanteau of the words Mediterranean and hurricanes) – can be very damaging because of the strong winds and the intense convective precipitations (thunderstorms) originating around the eye of the storm.

Medicane Apollo (named by a consortium of European meteorological services; see Meteoweb, 2021) formed on 28 October in the Ionian Sea, offshore of Sicily, from a very

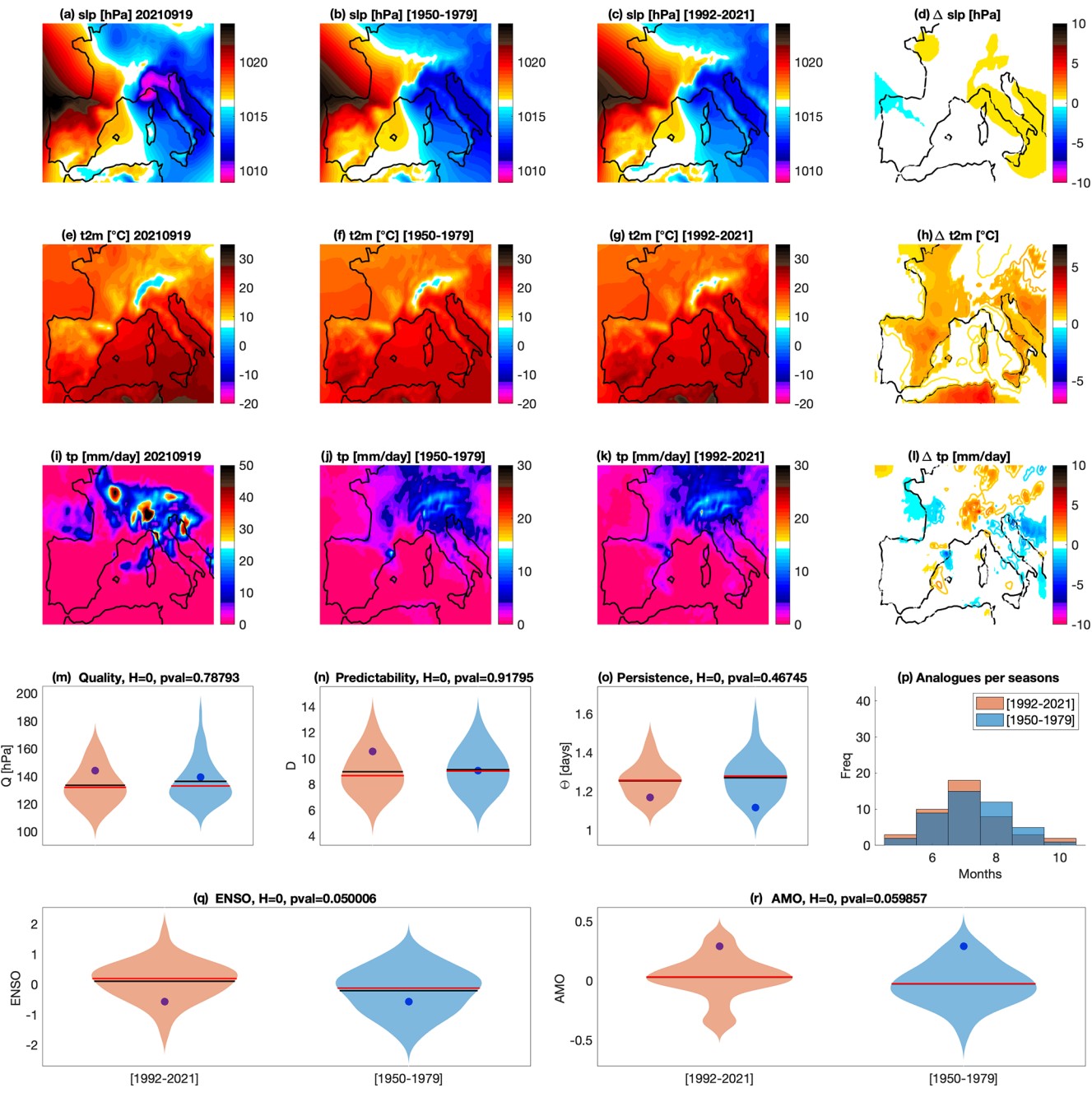

**Figure 8.** Attribution for the Po Valley tornado outbreak on 19 September 2021. Daily mean sea-level pressure slp (**a**), 2 m temperatures t2m (**e**) and total precipitation tp (**i**) on the day of the event. Average of the 33 sea-level pressure analogues found for the counterfactual [1950–1979] (**b**) and factual [1992–2021] (**c**) periods and corresponding 2 m temperatures (**f, g**) and daily precipitation rate (**j, k**). Δslp (**d**), Δt2m (**h**) and Δtp (**l**) between factual and counterfactual periods: colored–filled areas show significant anomalies with respect to the bootstrap procedure. Violin plots for counterfactual (blue) and factual (orange) periods for the analogue quality $Q$ (**m**), the predictability index $D$ (**n**), the persistence index $\Theta$ (**o**) and the distribution of analogues in each month (**p**). Violin plots for counterfactual (blue) and factual (orange) periods for ENSO (**q**) and AMO (**r**). Values for the peak day of the extreme event are marked by a blue dot. Horizontal bars in panels (**m**)–(**r**) correspond to the mean (black) and median (red) of the distributions.

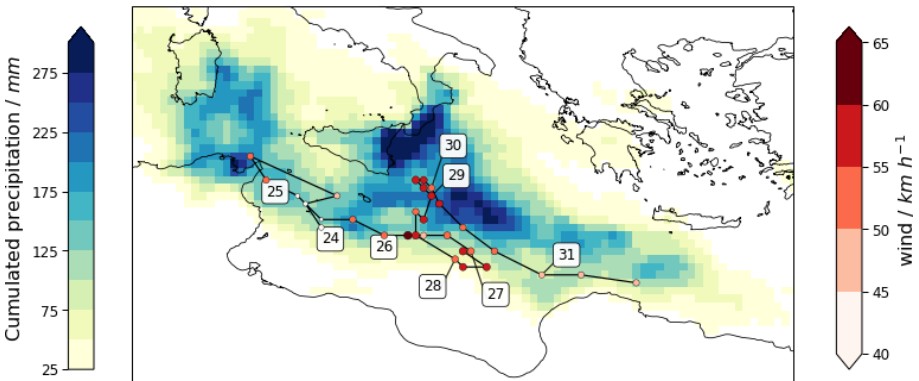

**Figure 9.** Track and associated precipitation for Apollo. Data from ERA5: track position is retrieved as the local minimum of slp, wind is the maximum wind speed in a 1.5° great-circle distance (GCD) radius of the slp center, precipitation is cumulated between 00:00 UTC on 24 October 2021 and 16:00 UTC on 31 October 2021. Time stamps indicate the first point for each day whose number is indicated.

active low-pressure disturbance. This low-pressure system was isolated near the Balearic Islands around 22 October and then moved to the central Mediterranean Sea, producing self-regenerating thunderstorms in the area of Catania on 24 October. These thunderstorms, which occurred before the extratropical cyclone became a medicane, already resulted in extremely heavy rain and floods in Catania ($> 400$ mm rain in 48 h, estimated by SIAS, 2021). Figure 9 displays the track of the cyclone along its life cycle. During the tropical phase of Apollo, according to the latest report available, at least 10 people were killed by the storm in Sicily, Malta, Algeria and Tunisia (jbarisk, 2021). The highest wind gusts were measured on 29 October ($104 \, \text{km} \, \text{h}^{-1}$), and the minimum pressure was estimated at 999 hPa. The Sicilian meteorological service (SIAS) measured $> 200$ mm convective precipitation associated with Apollo in the area of Syracuse on the same date. Apollo started to weaken on 30 October 2021 and made landfall near Bayda, Libya, a few days later.

### 4.7.1 Medicanes and climate change

It is difficult to study trends in frequency and intensity of medicanes under climate change. First of all, our knowledge of historical medicanes is very limited before the satellite era, and they are rare events with an estimated frequency of between 1 and 2 events per year (Cavicchia et al., 2014a). Medicane genesis is favored when an extratropical depression gets isolated from the polar jet stream. This cutoff becomes quasi-stationary on the Mediterranean Sea and can use the large availability of heat and humidity from the sea to produce organized convection. Recent studies of medicanes under climate change have therefore considered two elements: the precursors, namely the cutoff low, and the potential for organized convection once the first condition is met (Cavicchia et al., 2014b; Romero and Emanuel, 2017; Tous et al., 2016). On one hand, a recent study suggests that the jet stream will shift northward (Stendel et al., 2021) and

therefore cutoff lows on the Mediterranean Sea may become slightly less frequent. On the other hand, the Mediterranean Sea is warming faster than the larger oceans, increasing the potential for convection once a depression system is present in the area. We then expect to see fewer medicanes but more intense ones (González-Alemán et al., 2019).

### 4.7.2 Attribution of Medicane Apollo to climate change

We now use the ERA5 dataset to perform the attribution of the cyclonic circulation associated with Apollo in the past and present climates (Fig. 10). We note that we will select analogues independently of the extratropical or tropical nature of the depression that has produced them. The analogue average slp values for both the factual and counterfactual periods (Fig. 10b, c) do not reach slp minima comparable to that of Apollo (Fig. 10a), although this may partly be an effect of averaging maps with cyclones at slightly different locations. The $\Delta$slp (Fig. 10d) displays a weak yet significant positive anomaly over the northern part of the domain, indicating that factual analogue cyclones are less deep or southward-shifted relative to counterfactual ones. Furthermore, we observe that temperatures are significantly warmer in the factual world, especially on the island of Sicily and on the southern Mediterranean basin (Fig. 10e–h). This warming is associated with a significant increase in precipitation in the factual period, likely due to the larger availability of heat and humidity from the sea (Fig. 10i–l). These results must be interpreted with care because the analogue quality clearly shows that Apollo's circulation pattern is extremely rare compared with the rest of its analogues (Fig. 10m). Apollo thus appears to be a black swan event. We do not detect remarkable changes in the distributions of the predictability index $D$ (Fig. 10n) or the persistence $\Theta$ (Fig. 10o). However, the event itself displays a lower $D$ and higher $\Theta$ when these are computed using factual data rather than counterfactual data. This could also have contributed to enhancing the persistence

of precipitation on the same areas. We do see a clear increase in analogues in the month of September in the factual period (Fig. 10p): this is the warmest month for the Mediterranean Sea, hence the most favorable for the development of deep convection in association with cyclonic depressions. This factor can greatly enhance precipitation, especially on the mountain ranges exposed to the winds, as in the case of Apollo, for the Etna and the Peloritani mountain ranges in Sicily. Finally, no significant differences in ENSO (Fig. 10q) and AMO (Fig. 10r) distributions conditioned to analogues have been found between the factual and counterfactual periods.

In keeping with the general trends reported in Sect. 4.7.1, our analysis highlights the potential intensification of precipitation associated with cyclones around the island of Sicily, supported both by higher temperatures and increased occurrence of cyclones in the month of September, the warmest for the Mediterranean Sea. However, we point to the black swan nature of this storm compared to its analogues and therefore to a careful interpretation of the attribution results obtained above.

## 4.8 Scandinavian cold spell

During late November 2021, Scandinavia experienced record-low temperatures for the season. On 28 November, the Nikkaluokta weather station in Sweden recorded $-37.4\,°C$, which was the lowest November temperature recorded in the country since 1980. Other stations in northern Sweden recorded their lowest November temperatures since the 1950s (SMHI, 2022a). Comparable records occurred in the first days of December. In Norway, the $-36.7\,°C$ recorded in Kautokeino was the lowest November reading since 2002 (SMHI, 2022b). These frigid temperatures were part of a broader area of below-average temperatures, peaking in the last week of November and first days of December, and stretching from northwestern Russia all the way to Spain (which recorded one of the top 10 coldest November months on record, AEMET, 2022). The cold spell impacted transports, including suspension of entire train lines (SVT, 2022) and an unusually large number of road accidents in southern Sweden (SVD, 2022).

The cold spell was associated with a large ridge forming over the North Atlantic starting from 23 November and drawing cold Arctic and Siberian air over the continent. A pressure dipole with a high over Scandinavia and a low over central Europe further favored cold air advection. The Atlantic Ridge persisted until early December, after which a more zonal circulation occurred, bringing warmer air masses over large parts of Europe.

### 4.8.1 Scandinavian cold spells and climate change

As discussed in Sect. 4.2.1, it is virtually certain that there has been a decrease in severity and/or frequency of cold spells in the last several decades, and the consensus is that at a global level this decrease will continue in the future. Scandinavia fits this trend and has shown a significant decrease in wintertime cold days in recent decades (Matthes et al., 2015). In the future, the decrease in wintertime cold days is expected to be stronger than in several other European regions (Dosio, 2016), as is the increase in yearly minimum daily-mean temperature (Bernes, 2017, p. 102).

### 4.8.2 Attribution of the Scandinavian cold spell to climate change

Figure 11 shows the results of our attribution analysis for the Scandinavian cold spell. The slp analogues suggest that the pressure dipole over Europe seen during the cold spell is quite an unusual configuration and that such a dipole has typically become weaker in the factual period (Fig. 11a–d). The weaker dipole in the analogues during both periods corresponds to warmer t2m compared to the event, but there is only a weak increase in the temperatures of the analogues between the two periods over Scandinavia. The only exceptions are the coastal areas in western and northern Norway (Fig. 11h). There is additionally a strong increase in temperatures over the Norwegian and Greenland seas and northeastern Europe, in keeping with the lower pressure to the north of Scandinavia in the factual period compared to the counterfactual period (Fig. 11d). The lack of a significant warming signal across Scandinavia is coupled to modest changes in the seasonality of the analogues (Fig. 11p) and in precipitation and the associated cloudiness (Fig. 11l). We hypothesize that the cold Siberian air masses contributing to the low Scandinavian temperatures during these events may not have warmed significantly (Cohen et al., 2013).

The quality of the analogues is good and shows little change when moving from the counterfactual to the factual world (Fig. 11m), as does their predictability (Fig. 11n). Interestingly, the unusualness of the slp dipole configuration highlighted in Fig. 11a–c thus does not translate to an unusually poor analogue quality for the event in question. The persistence $\Theta$ of the analogue patterns shows a weak increase in the factual period, albeit with no significant change in the distribution; the persistence of the event itself computed on the factual data instead increases sharply relative to that computed on the counterfactual data (Fig. 11o). This provides an alternative hypothesis to explain the weak change in Scandinavian temperatures between the two periods, in addition to the above-discussed weak warming of Siberian air masses. Indeed the longer persistence of the slp pattern – and hence of the cold advection – in the factual period could partly compensate for the effect of warmer air being advected over the region. The increased persistence may be compared to the findings of Matthes et al. (2015), who find no shifts from longer to shorter cold spells in the northern high latitudes, except for a decrease in only the longest episodes.

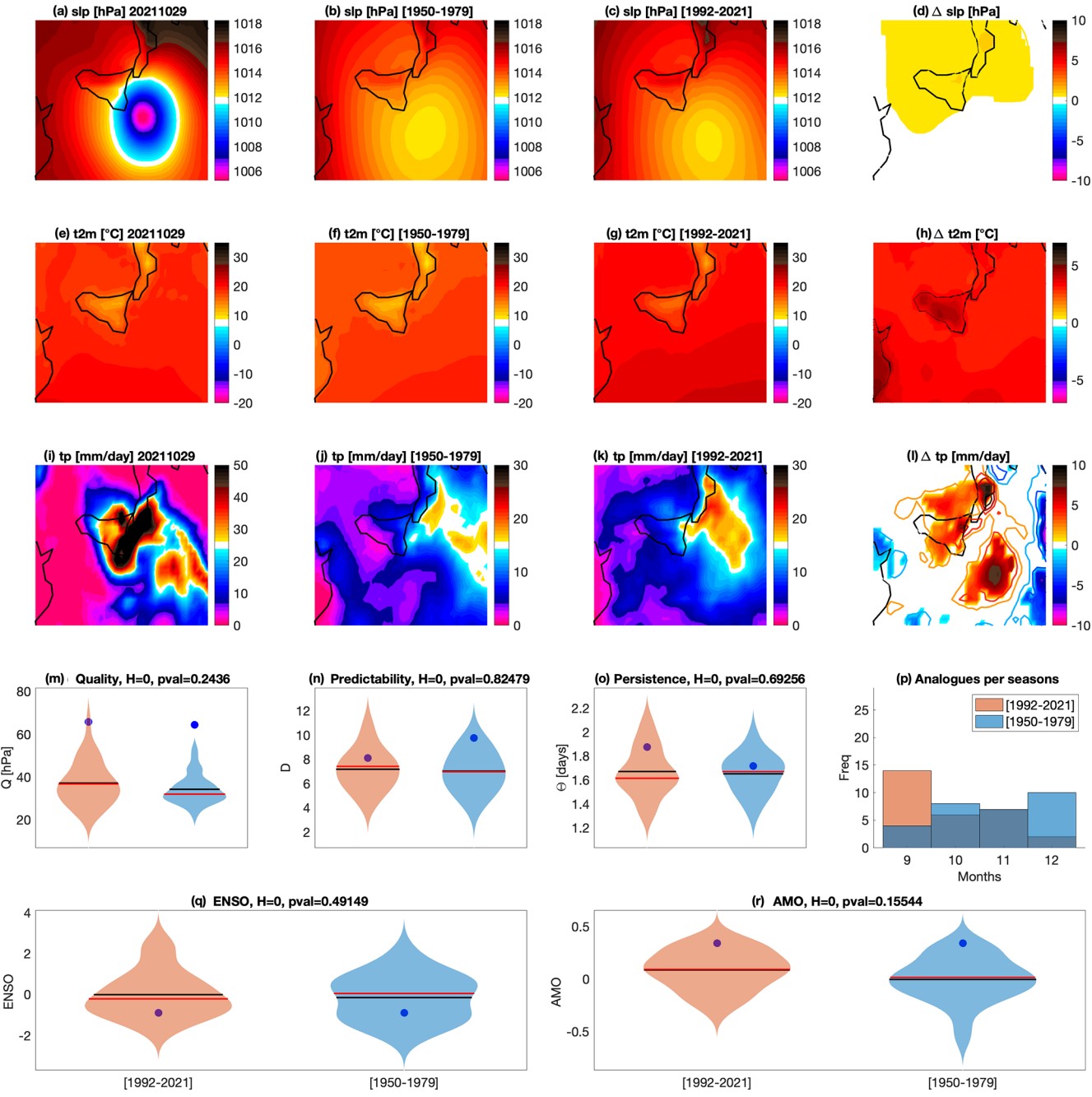

**Figure 10.** Attribution for the Medicane Apollo on 29 October 2021. Daily mean sea-level pressure slp **(a)**, 2 m temperatures t2m **(e)** and total precipitation tp **(i)** on the day of the event. Average of the 33 sea-level pressure analogues found for the counterfactual [1950–1979] **(b)** and factual [1992–2021] **(c)** periods and corresponding 2 m temperatures **(f, g)** and daily precipitation rate **(j, k)**. Δslp **(d)**, Δt2m **(h)** and Δtp **(l)** between factual and counterfactual periods: colored–filled areas show significant anomalies with respect to the bootstrap procedure. Violin plots for counterfactual (blue) and factual (orange) periods for the analogue quality $Q$ **(m)**, the predictability index $D$ **(n)**, the persistence index $\Theta$ **(o)** and the distribution of analogues in each month **(p)**. Violin plots for counterfactual (blue) and factual (orange) periods for ENSO **(q)** and AMO **(r)**. Values for the peak day of the extreme event are marked by a blue dot. Horizontal bars in panels **(m)**–**(r)** correspond to the mean (black) and median (red) of the distributions.

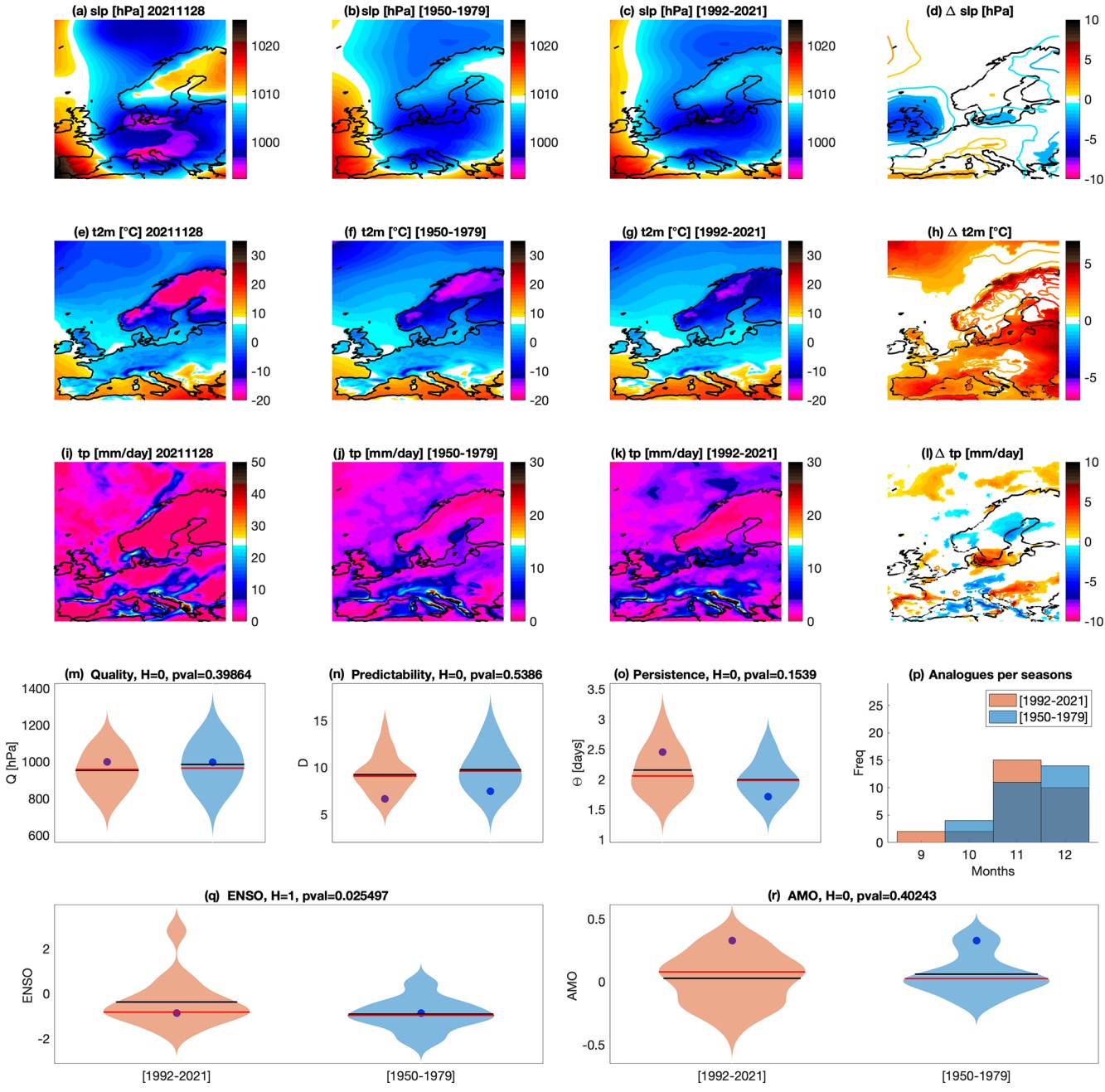

**Figure 11.** Attribution for the Scandinavian cold spell on 28 November 2021. Daily mean sea-level pressure slp **(a)**, 2 m temperatures t2m **(e)** and total precipitation tp **(i)** on the day of the event. Average of the 33 sea-level pressure analogues found for the counterfactual [1950–1979] **(b)** and factual [1992–2021] **(c)** periods and corresponding 2 m temperatures **(f, g)** and daily precipitation rate **(j, k)**. Δslp **(d)**, Δt2m **(h)** and Δtp **(l)** between factual and counterfactual periods: colored–filled areas show significant anomalies with respect to the bootstrap procedure. Violin plots for counterfactual (blue) and factual (orange) periods for the analogue quality $Q$ **(m)**, the predictability index $D$ **(n)**, the persistence index $\Theta$ **(o)** and the distribution of analogues in each month **(p)**. Violin plots for counterfactual (blue) and factual (orange) periods for ENSO **(q)** and AMO **(r)**. Values for the peak day of the extreme event are marked by a blue dot. Horizontal bars in panels **(m)**–**(r)** correspond to the mean (black) and median (red) of the distributions.

Finally, the analogues show a significant change in ENSO distribution, with a weak shift towards more positive ENSO phases in the factual period (Fig. 11q). There is an association between positive ENSO and more severe European winters (e.g., Fraedrich, 1990, 1994), which may provide a further explanation for the lack of significant warming in the factual period analogues. The changes in AMO distribution associated with the analogues in the two periods are inconclusive (Fig. 11r).

Based on the above, we conclude that the atmospheric configuration driving cold spells such as the November 2021 episode has not become more unusual with climate change and that the intensity of the cold spells engendered by similar atmospheric configurations has not weakened significantly, contrary to the decreasing trends observed in data and model simulations for cold days in Scandinavia (Sect. 4.8.1). ENSO may provide some modulation of the cold spell characteristics between the two periods, but based on the moderate changes observed in its association with the cold spell analogues, we deem it unlikely to be the main physical driver of our results. We thus interpret the November 2021 event as a persistent cold extreme in a warming climate.

# 5  Conclusions

We have analyzed the atmospheric circulation associated with a selection of high-impact extreme events occurring in 2021 from an attribution perspective. Specifically, we have performed a semi-objective selection of a representative circulation pattern for each extreme and have then identified two sets of analogues: the first in the 1950–1979 period, which approximates a counterfactual world; the second in the 1992–2021 period, which approximates a factual world. Regardless the specificity of each event, our analysis evidences the relevant role of atmospheric circulation changes under anthropogenic climate change in controlling the characteristics of many of these events, which is a conclusion of relevance to the broader field of extreme-event attribution.

A second important outcome of this study is to include, in the attribution framework, the systematic use of the dynamical indicators of persistence and predictability. Persistence is of particular interest, since there has been a lively scientific debate on changes in atmospheric persistence and how these may affect extreme events (Coumou and Rahmstorf, 2012; Hoskins and Woollings, 2015; Wehrli et al., 2020).

Finally, we have studied the quality of the analogues – namely the typicality of the analogues relative to the atmospheric variability – and their changes over time. This brings a third relevant outcome, namely the ability to understand whether both a given circulation and its analogues are becoming more or less typical (i.e., have better or worse analogues). The two do not always vary in tandem, meaning that the quality of the analogues for a given extreme may remain unchanged while the analogues of the analogues become better. While not immediate to interpret, this provides some subtle insights into how the configurations conducive to an extreme relate to the broader atmospheric variability typical of a given climate. In the case of Medicane Apollo, the lack of good-quality analogues directly points to the unprecedented nature of this event, making it a black swan among the weather patterns in Europe. It is therefore questionable to attempt any attribution statements in this case. This finding is also a warning that weather extreme events do not necessarily belong to the sample of weather situations observed in the last several decades.

The main limitations of our framework include the somewhat arbitrary choice of the region used to define the analogues, the timescale for the selection of the analogues and the number of analogues retained for analysis. Moreover, only for two of the events – the Po Valley tornado outbreak and Medicane Apollo – is it possible to statistically exclude a role of natural inter-decadal variability of ENSO and AMO in explaining differences in the analogues. We are well aware of these limitations and have designed the study to minimize their impact. The main advantage of working with analogues of sea-level pressure is the possibility of applying expert judgment to select a region that includes the large-scale cyclonic/anticyclonic structures concurring with the event. The use of daily means allows us to average out the daily cycle. Longer timescales have been tested, but they produce worse analogues due to the fact that the synoptic structures move too much and lead to aliased atmospheric patterns. We nonetheless believe that longer timescales could be used to study long-lasting extreme events such as droughts. Furthermore, at daily time resolution information about the stationarity or lack thereof of the patterns is retained in the persistence metric. We have tested the dependence of our results on the number of analogues used and found that numbers between 30 and 60 analogues provide a good balance between having meaningful statistics and selecting good-quality analogues. Finally, we highlight that conventional extreme value attribution shares many of the same limitations, including the choice of the region, thresholds and timescale.

Our approach does not want to substitute extreme-event attributions based on the statistical fitting of extreme value distributions: those approaches can be used to inform stakeholders of changes in return times of extreme events in factual versus counterfactual worlds, and they have been successfully used by the attribution community in a large number of instances (Trenberth et al., 2015; Van Oldenborgh and Van Ulden, 2003; Vautard and Yiou, 2012; Van Oldenborgh et al., 2012; Trenberth et al., 2015; Vautard et al., 2016, 2018). We rather see our analysis as complementing statistical approaches by providing insights on the possible changes over time of the dynamics underlying specific extreme events, as described by National Academies of Sciences, Engineering, and Medicine (2016). Further development of this methodology can include the use of analogues to flag populations of events that share the same dynamical

origin, on the line of research proposed by Jézéquel et al. (2018b) and Shepherd (2019). This would allow us to use the tools of statistical attribution with an additional conditioning from the analogues and to release an automated package that produces these analyses in a matter of minutes as soon as the ERA5 data are available. Other possible extensions include searching for analogues of different observables such as geopotential height, temperature on pressure levels, winds and more. Although valuable, these options must be evaluated with extreme care in the context of attribution because of the non-linear trends already introduced by the anthropogenic forcing on the average of these quantities (Jézéquel et al., 2018a).

To conclude, the analogue approach to extreme-event attribution shows that many extreme events are significantly modified in the present climate with respect to the past, because of changes in the position, persistence and seasonality of cyclonic/anticyclonic patterns. Our approach, complementary to the statistical methods already available in the attribution community, underscores the importance of considering changes in the atmospheric circulation when performing attribution studies.

## Appendix A:  Predictability and persistence indices

The attractor of a dynamical system is a geometric object defined in the space hosting all the possible states of the system (phase space). Each point $\zeta$ on the attractor can be characterized by two dynamical indicators: the local dimension $D$, which indicates the number of degrees of freedom active locally around $\zeta$, and the persistence $\Theta$, a measure of the mean residence time of the system around $\zeta$ (Faranda et al., 2017). To determine $D$, we exploit recent results from the application of extreme value theory to Poincaré recurrences in dynamical systems. This approach considers long trajectories of a system – in our case successions of daily slp latitude–longitude maps – corresponding to a sequence of states on the attractor. For a given point $\zeta$ in phase space (e.g., a given slp map), we compute the probability that the system returns within a ball of radius $\epsilon$ centered on the point $\zeta$. The Freitas et al. (2010) theorem, modified by Lucarini et al. (2012), states that logarithmic returns,

$$g(x(t)) = -\log(\text{dist}(x(t), \zeta)),  \tag{A1}$$

yield a probability distribution such that

$$\Pr(z > s(q)) \simeq \exp\left[-\vartheta(\zeta)\left(\frac{z - \mu(\zeta)}{\sigma(\zeta)}\right)\right],  \tag{A2}$$

where $z = g(x(t))$ and $s$ is a high threshold associated with a quantile $q$ of the series $g(x(t))$. Requiring that the orbit falls within a ball of radius $\epsilon$ around the point $\zeta$ is equivalent to asking that the series $g(x(t))$ is over the threshold $s$; therefore, the ball radius $\epsilon$ is simply $e^{-s(q)}$. The resulting distribution is the exponential member of the generalized Pareto

distribution family. The parameters $\mu$ and $\sigma$, namely the location and the scale parameter of the distribution, depend on the point $\zeta$ in phase space. $\mu(\zeta)$ corresponds to the threshold $s(q)$, while the local dimension $D(\zeta)$ can be obtained via the relation $\sigma = 1/D(\zeta)$. This is the metric of predictability introduced in Sect. 3.

When $x(t)$ contains all the variables of the system, the estimation of $D$ based on extreme value theory has a number of advantages over traditional methods (e.g., the box counting algorithm, Liebovitch and Toth, 1989; Sarkar and Chaudhuri, 1994). First, it does not require us to estimate the volume of different sets in scale space: the selection of $s(q)$ based on the quantile provides a selection of different scales $s$ which depends on the recurrence rate around the point $\zeta$. Moreover, it does not require the a priori selection of the maximum embedding dimension as the observable $g$ is always a univariate time series.

The persistence of the state $\zeta$ is measured via the extremal index $0 < \vartheta(\zeta) < 1$, a dimensionless parameter, from which we extract $\Theta(\zeta) = \Delta t / \vartheta(\zeta)$. Here, $\Delta t$ is the time step of the dataset being analyzed. $\Theta(\zeta)$ is therefore the average residence time of trajectories around $\zeta$, namely the metric of persistence introduced in Sect. 3, and it has units of time (in this study days). If $\zeta$ is a fixed point of the attractor, then $\Theta(\zeta) = \infty$. For a trajectory that leaves the neighborhood of $\zeta$ at the next time iteration, $\Theta = 1$. To estimate $\vartheta$, we adopt the Süveges estimator (Süveges, 2007). For further details on the extremal index, see Moloney et al. (2019).

*Code availability.* The code to compute the dynamical indicators of predictability $D$ and persistence $\theta$ is available at https://fr.mathworks.com/matlabcentral/fileexchange/95768-attractor-local-dimension-and-local-persistence-computation (Faranda, 2022).

*Data availability.* ERA5 reanalysis is available from the Copernicus Climate Change Service Climate Data Store (https://doi.org/10.24381/cds.bd0915c6, Hersbach et al., 2018).

*Author contributions.*  DF conceived the study, performed the attribution analysis for all events and wrote the section on the Medicane Apollo. SB wrote the section on Hurricane Ida and performed cyclone tracking analyses. MG wrote the section about storm Filomena. MK wrote the section about the Mediterranean heat wave. RN wrote the section about the Westphalia floods. FP wrote the section about the Po Valley tornado outbreak. PY wrote the section about the French spring cold spell. GM wrote the section about the Scandinavian cold spell and contributed to the final drafting of the manuscript. All the authors contributed to writing and reviewing the introduction, methods, and conclusions of the article.

*Competing interests.* The contact author has declared that none of the authors has any competing interests.

*Acknowledgements.* The authors wish to thank Maria del Carmen Alvarez-Castro, Jacopo Riboldi, Melinda Galfi, Mathieu Vrac, Andreia Hisi, Erika Coppola, Robert Vautard and the two anonymous reviewers for useful discussions and inputs.

*Financial support.* This research has been supported by the Agence Nationale de la Recherche (grant nos. ANR-19-ERC7-0003 (BOREAS) and ANR-20-CE01-0008-01 (SAMPRACE)), the Centre National de la Recherche Scientifique (MANU project DINCLIC), and the European Union's Horizon 2020 research and innovation program (grant no. 101003469 (XAIDA), European Research Council (ERC) grant no. 948309 (CENÆ), and Marie Skłodowska-Curie grant no. 956396 (EDIPI)).

*Review statement.* This paper was edited by Peter Knippertz and reviewed by two anonymous referees.

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
