# Peer review of "A climate-change attribution retrospective of some impactful weather extremes of 2021"

_Weather and Climate Dynamics, 2022_

## Author Response (AR1)

**Dear Editor,**

**We are pleased to resubmit a new version of our manuscript for publication in WCD. We have taken into the maximum considerations the concerns raised by the reviewers and performed the demanded changes which have, in our opinion, greatly improved the manuscript. While a detailed answer for each of the comments is provided below, we would like to underline here few main features of the new version:**

- **A better framing of the existing literature including careful quotations of the IPCC report, an overview of the existing methods and a critical assessment of the limitations of our study.**
- **Rigorous statistical tests to highlight significance differences in factual and counterfactual analogues distributions.**
- **A focus on seasonal analogues to avoid mixing different physical processes modulating extremes in different periods of the year.**
- **An assessment of the role of natural variability through the analysis of ENSO and AMO indices.**

**We hope that this version of the manuscript will be suitable for publication in WCD.**

**Best wishes,**
**Davide Faranda,**
**on behalf of the authors**

**Answers Reviewer 1:**

Overall comments

This paper uses atmospheric circulation analogues to study eight extreme weather events in 2021 which occurred mainly over Europe (with one over the eastern USA), asking how they may have been affected by anthropogenic climate change, and in particular, by forced changes in the properties of the circulation regimes --- which is to say, changes in dynamics. The question is topical, and interesting. The attribution of changes in extreme weather, and predictions of future changes in extremes, are invariably based on thermodynamics, because that is the aspect of climate change in which there is high confidence. Yet extreme weather events invariably involve particular dynamical conditions, implying that changes in those dynamical conditions could potentially be part of the response to anthropogenic climate change. The difficulty is that any such changes are far more uncertain than the thermodynamic changes, as has been widely discussed in the literature. The contrast in levels of confidence is most apparent in the statements issued by the IPCC on this matter. What this paper does, then, is to assess the relative contribution of changes in thermodynamics (by which I mean the difference in hazard conditional on the circulation analogue) and changes in dynamics (by which I mean the changes in properties of the circulation analogues), partly empirically (by identifying changes in the ERA5 reanalysis between 1950-1979 and 1992-2021), and partly theoretically (by drawing on existing literature, especially the AR6 report). It does this for a wide range of different kinds of extreme weather events. The synoptic descriptions of the events are very nice, and the format of the paper is potentially an interesting one. The problem is that the methodology is

highly problematical, and the conclusions drawn are in many cases far too definitive. I am also absolutely shocked by how the conclusions of the AR6 have been twisted by the authors in a highly misleading fashion. Thus, the paper is completely unpublishable in its present form.

I briefly summarize the conclusions of the eight case studies (the justification for my statements is provided in my detailed comments):

Winter storm Filomena: unsubstantiated claim

French Spring cold spell: the expected result from thermodynamics

Westphalia Floods: unsubstantiated claim

Mediterranean summer heatwave: the expected result from thermodynamics

Hurricane Ida: the expected result from thermodynamics

Po Valley tornadoes outbreak: inconclusive (as in the IPCC for tornadoes in general)

Medicane Apollo: unsubstantiated claim

Late autumn Scandinavian cold-spell: such cold extremes do not warm

With the exception of the last one, which seems potentially interesting, the attribution statements are either what one would expect from the IPCC reports, or they involve unsubstantiated claims. That would not seem to provide much basis for a publication. However, I can imagine a reframing of the results which could be publishable. For cases where there are good analogues, no apparent changes in those analogues between 1950-1979 and 1992-2021, and a thermodynamic effect which is consistent with physical understanding, the analysis supports a purely thermodynamic attribution of the event, allowing a strong statement to be made. That is interesting in itself. For cases where there are good analogues and apparent changes in those analogues between 1950-1979 and 1992-2021, the analysis raises questions about potential changes in dynamical conditions in response to climate change. One cannot simply assume, as is done here, that any such changes are anthropogenic; but one can articulate various hypotheses, based on the literature. That would improve the quality of extreme event attribution and risk assessment. And for cases where there are no good analogues, as is found here for Filomena and Apollo, the identification of the events as 'black swans' is very important as it suggests that a standard statistical attribution would be completely unjustified. This would also improve the quality of extreme event attribution, as well as stimulating research into how to treat such singular events.

Thus, I would suggest building more explicitly from the synoptic descriptions of the events, which I found to be much along the lines of the wonderful study of Black et al. (2004 Weather) on the 2003 European heat wave, and then articulate the contribution of thermodynamic and of plausible dynamical responses to climate change, being careful to avoid categorical language in the latter case (see detailed comments below). This could be done without a drastic overhaul of the manuscript, hence my recommendation of major revision rather than rejection. However, I need to emphasize quite firmly that many of the problems I identify below are fatal, in my view, if not addressed.

**We thank the reviewer for reading our manuscript in detail. We appreciate that the reviewer recognizes the importance of attribution studies focusing on the dynamics and that they enjoyed the synoptic description of the events.  We do understand the criticisms expressed and we have reviewed the manuscript following their suggestions. We have  addressed  all comments by a substantial reformulation of the paper, both when it comes to the discussion of the existing literature and the interpretation of the results. In particular we have focussed the discussion by clearly**

**pointing out the different dynamical and thermodynamic drivers of the extreme events, or those who suffer from the lack of good analogues (black swans). This has implied a rewriting of the interpretation part of these events. Extreme care has been taken to avoid confusion and misinterpretation of the IPCC AR6 report: we have made a clear separation between IPCC statements (that have been reported within quotes) and other results available in the scientific literature. We have also relaxed the assumption that the changes we identify are necessarily anthropogenically driven, better describing our hypothesis about taking 30 years of factual and counterfactual worlds, as is done in many attribution studies (see, e.g. Vautard et al. 2016, Paciorek et al. 2018, Van Oldenborgh et al. 2019) . Furthermore, we have devised a statistical procedure to control for the effect of lower-frequency and inter-decadal variability, such as that caused, for example, by the Atlantic Multi-Decadal Oscillation or by low-frequency modulations of the El Nino--Southern Oscillation. If a direct influence of such low-frequency variability can be excluded, then changes in analogues between the two periods we consider can be attributed to the climate change signal.**

**Vautard, R., Yiou, P., Otto, F., Stott, P., Christidis, N., Van Oldenborgh, G. J., & Schaller, N. (2016). Attribution of human-induced dynamical and thermodynamical contributions in extreme weather events. *Environmental Research Letters*, *11*(11), 114009.**

**Paciorek, C. J., Stone, D. A., & Wehner, M. F. (2018). Quantifying statistical uncertainty in the attribution of human influence on severe weather. *Weather and climate extremes*, *20*, 69-80.**

**Van Oldenborgh, G. J., Philip, S., Kew, S., Vautard, R., Boucher, O., Otto, F., ... & van Aalst, M. (2019). Human contribution to the record-breaking June 2019 heat wave in France. *World Weather Attribution*.**

Detailed comments

My jaw dropped when reading the very first paragraph of the paper (lines 19-29), which I regard as a complete misrepresentation of the conclusions of the AR6 WGI report. The SPM is cited in support of the statement that "anthropogenic climate change is critically affecting the dynamics of weather extremes" (line 20). When I read the SPM for statements about extremes, I can find no statement that could remotely be construed as supporting any conclusion about attribution of changes in the dynamics of extremes to anthropogenic climate change. (If I missed something, I would be happy to be corrected on this point.) On the contrary, all SPM statements about changes in extremes would appear to be anchored in thermodynamics. Even if the length of heat waves is increasing, this does not imply a change in dynamics; it is simply that if the mean temperature increases, then the mean time of exceedance above a fixed temperature threshold will necessarily increase (all else being equal). I found two statements in the SPM concerning potential changes in dynamics:
"B.3.2 A warmer climate will intensify very wet and very dry weather and climate events and seasons, with implications for flooding or drought (high confidence), but the location and frequency of these events depend on projected changes in regional atmospheric circulation, including monsoons and mid-latitude storm tracks."
"C.1.3 Internal variability has largely been responsible for the amplification and attenuation of the observed human-caused decadal-to-multi-decadal mean precipitation changes in many land regions (high confidence)."

The first statement says that future changes could depend on dynamics, which is a truism, but is certainly not implying any kind of attribution or definitive knowledge. The second statement is suggesting that dynamical modulation of the thermodynamic changes seen so far can be mainly attributed to internal variability, not anthropogenic climate change. Thus, the AR6 SPM is telling us that any forced changes in the dynamics of extremes are highly uncertain, and that any observed changes are dominated by internal variability.

I also could find no statement in the Executive Summary of AR6 Chapter 11 (on extremes) that could remotely be construed as supporting any conclusion about attribution of changes in the dynamics of extremes to anthropogenic climate change.

**We accept the criticism that we have not been  precise in quoting the AR6 report here, and we should have also referred to the full report or the technical summary rather than the SPM only. However, we want to stress to the reviewer, the editorial board and the readers of WCD that our intention was not to "twist" the statements of the IPCC Report. We have revised the paper according to the reviewer's suggestion and tone down our statements. In particular, the sentence : "anthropogenic climate change is critically affecting the dynamics of weather extremes" has been substituted by "anthropogenic climate change is critically affecting weather extremes". Indeed, the nature of these changes is mostly thermodynamics, and the IPCC report provides limited evidence that potential changes in extreme events dynamics could be due to anthropogenic changes. As quoted by the Reviewer, the SPM report states that: "B.3.2 A warmer climate will intensify very wet and very dry weather and climate events and seasons, with implications for flooding or drought (high confidence), but the location and frequency of these events depend on projected changes in regional atmospheric circulation, including monsoons and mid-latitude storm tracks."  We hope that the reviewer will enjoy the way we cite the IPCC report in the manuscript and the net separation with other studies that are (yet) not acknowledged in the report.**

In lines 20-23, this paper states "For summer, the AR6 report states that we are already observing prolonged periods of extremely warm conditions (Horton et al., 2016) with increased droughts leading to forest fires (Flannigan et al., 2000), species extinctions (Román-Palacios and Wiens, 2020) and health issues for vulnerable populations (Mitchell et al., 2016)."

As an elaboration of the preceding statement about changes in the dynamics of extremes, and as a characterization of the AR6, this is completely misleading. Horton et al. (2016) discusses the potential for changes in dynamics to affect the nature of heat extremes, but makes clear that any such changes are highly uncertain and controversial, and in any case this paper is not cited by AR6 Chapter 11. Perhaps the authors meant Horton et al. (2015), which is cited by AR6 Chapter 11, but that paper is about observed trends and makes no attribution, as would be implied by the word "already". As noted above, prolonged periods of warm conditions are a straightforward consequence of mean warming and do not require a dynamical explanation. The statements about impacts are similarly anchored in thermodynamic mechanisms.

**We have  followed  the suggestion of the reviewer by rephrasing accordingly the introduction (LL20-L26)**

In lines 23-24, this paper states "In winter, increased persistence of cyclonic and anticyclonic structures leads to extremely wet and dry periods (Ogawa et al., 2018)"
The wording suggests that such a conclusion can be found in the AR6, but I could find no reference to Ogawa et al. (2018) in Chapter 11, and as noted earlier, no attribution of changes in the dynamics of extremes in the SPM or the Executive Summary of Chapter 11.

**We were unclear in this passage, and indeed did not intend to refer to the AR6 any longer. In the revised text we have clarified that we are citing other relevant scientific literature, not necessarily linked to AR6. We have rephrased this sentence as: " Recent scientific  literature points to the need of understanding the role of dynamical drivers of changes in weather extremes: in winter, increased persistence of cyclonic and anticyclonic structures can lead to extremely wet or dry periods (Berkovic and Raveh-Rubin, 2022) on the Eastern Mediterranean. Such change in persistence of synoptic structure is also expected to change with global warming in the northern hemisphere summer( see, e.g. Kornhuber and Tamarin-Brodsky (2021)). Gordon et al. (2005); Bala et al. (2010) and Pendergrass et al. (2017) suggest that, in the shoulder seasons, we observe a large variability of rains associated with both tropical and extratropical storms and convective events, leading to an alteration of the hydrological cycle (LL 28-35).**

In lines 25-27, this paper states "Finally, the IPCC also warns that, in the shoulder seasons, we observe a large variability of rains associated with both tropical and extratropical storms and convective events, leading to an alteration of the hydrological cycle (Gordon et al., 2005; Bala et al., 2010; Pendergrass et al., 2017)."I could find no support for this statement either in Chapter 11 or in Chapter 8 (on the hydrological cycle). There is an argument made for an increasing number of dry days in many regions, but again the argument is based on thermodynamics/energetics.

**Thank you, this has all been rewritten.  We invite the reviewer to read our modifications  LL 28-35**

Finally, in lines 27-29, this paper states "These trends are expected to accelerate in the coming years, if the global efforts to reduce carbon emissions are not implemented swiftly (Trisos et al., 2020)." Trisos et al. (2020) is about biodiversity loss and while I haven't read the paper, I would be surprised if it was not based on the sort of thermodynamic arguments for increased hazard represented in the AR6. It seems highly misleading to use that reference the way it is used here.

**We have removed the quoted sentence.**

I must admit that I am at something of a loss when I read the first paragraph of a paper and find that every single sentence is highly misleading and a severe distortion of the literature. In fact, I don't believe that I have ever had that experience before. However, this paragraph is only intended to be motivational, and I have no disagreement with what is said in the rest of p.2, or the subsequent motivation for this study. Thus, while the opening paragraph definitely needs a complete overhaul, I will press on with my review.

**We hope that the previous answers fully address the concerns of the reviewer, provide a truer account of the IPCC AR6 statements and clearly separate these from the rest of the literature discussed in the introduction.**

Lines 53-57: This wording appears to be suggesting that the authors consider any change in any atmospheric statistic between 1950-1979 and 1992-2021 to be the forced response to anthropogenic climate change. Quite apart from the potential inhomogeneity issue arising from comparing reanalyses prior to 1980 (before the satellite era) and after 1980 – which is always a serious concern, and needs to be addressed here – this approach seems far too liberal, and out of step with both the scientific understanding of multi-decadal variability (which is reflected, e.g., in erratic multi-decadal trends in the NAO), and the IPCC D&A framework which requires agreement between the spatial fingerprints of an observed trend and a well-accepted prediction from climate models in order to make any such attribution.

**We have understood and seriously addressed the concern of the reviewer.  The homogeneity issue is well known to affect essentially high latitudes and tropical regions (e.g., G. Sturaro, A closer look at the climatological discontinuities present in the NCEP/NCAR reanalysis temperature due to the introduction of satellite data, Climate dynamics 21 (2003) 309–316.). However, as a check of our assumption we have computed the Quality of the analogs (Q) for all days in the factual and counterfactual periods on the wide North Atlantic domain [80W-50E and 22.5N-70N] (see figure below) and applied a two-sided Cramér-von Mises test at the 0.05 significance level. The pvalue found (0.1995) imply that the null hypothesis that the two samples come from the same distribution cannot be rejected, hence supporting or claiming homogeneity. This has been added to LL123-128 of the manuscript**

[Figure]

*Figure R1: histogram of analogs qualities Q for all the day in the counterfactual and factual period on the wide North Atlantic domain [80W-50E and 22.5N-70N] .*

Furthermore,  we have used the latest release of ERA5 data with the back extension to 1959.  The reason why our approach has a negligible sensitivity to this homogeneity problem can  be explained with the fact that we compare atmospheric patterns, not pointwise values.

Regarding the claim that our approach to extreme events attribution is liberal, we would like to remark that it is in line with  the guidelines of the National Academy of Sciences "Attribution of Extreme Weather Events in the Context of Climate Change" rather than the detection and attribution approach outlined in the IPCC reports (see discussion in LL664-668).

Finally, in order to **control for the effect of lower-frequency and inter-decadal variability, such as that caused, for example, by the Atlantic Multi-Decadal Oscillation or by low-frequency modulations of the El Nino--Southern Oscillation. If a direct influence of such low-frequency variability can be excluded, then changes in analogues between the two periods we consider can be attributed to the climate change signal.  The rationale for this analysis is detailed in LL 146-156**

National Academies of Sciences, Division on Earth and  Life Studies, Board on Atmospheric Sciences and Climate, & Committee on Extreme Weather Events and Climate Change Attribution. (2016). *Attribution of extreme weather events in the context of climate change*. National Academies Press. DOI: 10.17226/21852

Lines 64-65: How do you account for the large-scale changes in slp associated with the thermodynamic effects of climate change, which presumably don't affect the circulation (since for circulation it is the horizontal gradients that matter), but would affect the Euclidean distances? Note that Chapter 10 of the IPCC AR4 report had a strong statement about increases in the strength of extratropical cyclones from climate change, largely based on the single study of Lambert & Fyfe (2006); it was subsequently recognized that taking minimum slp as a metric for extratropical cyclone intensity was fallacious as it was subject to the confounding influence of large-scale slp changes, and the AR5 had to row back on this statement. How can you convince the reader that you are not prone to the same problem? This might particularly affect the persistence metric.

**The rationale for using the sea-level pressure (and we note here that we use the entire map and not only the sea-level pressure minima) is that this observable is less subject to long term trends induced by the thermodynamic warming than, for example, Z500. Although the Lambert & Fyfe paper is interesting, it is largely outdated (AR4: horizontal resolution of 400km) and only discusses model simulations. We could not find any more recent study that mentions what the reviewer states, namely that " taking minimum slp as a metric for extratropical cyclone intensity was fallacious as it was subject to the confounding influence of large-scale slp changes". In the ERA5 data the horizontal resolution is 0.25°~ 30 km, an order of magnitude higher than in the AR4 simulations. This yields well-identifiable cyclones cores. Furthermore, we base our choice on the recent review of Walker (2020) who clearly states that " The most frequent choice is to use either local minima in MSLP or maxima in vorticity at a single geopotential height or pressure level (in the mid–lower troposphere) to identify an ETC and track that feature through time and space". Regarding the possibility that the SLP patterns are affected by changes in persistence, we outline that, in Faranda et al. (2019 Nature Communications), we have analyzed several sea-level pressure maps issued from a large sample of Reanalyses, CMIP5 historical simulations and future emission scenarios and found no trend in the persistence metric and modest trend in the dimension $d$ in the period considered in the present study. This is now accounted in LL76-79 of the manuscript**

**Faranda, D., Alvarez-Castro, M. C., Messori, G., Rodrigues, D., & Yiou, P. (2019). The hammam effect or how a warm ocean enhances large scale atmospheric predictability. Nature communications, 10(1), 1-7.**

**Walker, E., Mitchell, D., & Seviour, W. (2020). The numerous approaches to tracking extratropical cyclones and the challenges they present. Weather, 75(11), 336-341.**

Lines 146-148: This very strong statement about changes in the wintertime North Atlantic storm track, attributed to Chapter 4 of AR6, again seems highly misleading. What the Executive Summary of Chapter 4 actually says on this subject is this: "Substantial uncertainty and thus low confidence remain in projecting regional changes in Northern Hemisphere jet streams and storm tracks, especially for the North Atlantic basin in winter; this is due to large natural internal variability, the competing effects of projected upper- and lower-tropospheric temperature gradient changes, and new evidence of weaknesses in simulating past variations in North Atlantic atmospheric circulation on seasonal-to-decadal

timescales." There is a big difference between what the CMIP models might show, and what there is confidence in.

**We agree and we have followed the suggestion of the reviewer by rephrasing as: "The IPCC report highlights that "the number of extratropical cyclones (ETC) composing the storm tracks is projected to weakly decline in future projections, but by no more than a few percent change" and that "the reduction is mostly located on the equatorward flank of the storm tracks" (lee 2021 et al). However, it also states that: "substantial uncertainty and thus low confidence remain in projecting regional changes in Northern Hemisphere jet streams and storm tracks, especially for the North Atlantic basin in winter". Nonetheless, "despite small changes in the dynamical intensity of ETCs, there is high confidence that precipitation associated with ETCs will increase in the future" (LL185-190)**

Lines 150-151 say "According to Seneviratne et al. (2021), the number of ECT (sic) associated with strong winds over the North Atlantic and Europe will decrease." But what Chapter 11 of AR6 actually says is this: "There is low confidence in past changes of maximum wind speeds and other measures of dynamical intensity of extratropical cyclones. Future wind speed changes are expected to be small, although poleward shifts in the storm tracks could lead to substantial changes in extreme wind speeds in some regions (medium confidence)." How can this text from Chapter 11 of AR6 possibly be twisted into the highly misleading and very categorical statement made by the authors? The AR6 surely appreciates that the most intense ETCs could potentially strengthen because of more latent heat release, and the current generation of CMIP models are far from being able to give a definitive answer on this.

**Thank you for the suggestion, this part has been rewritten, see LL185-195 in the new version of the manuscript.**

As a result, the statement made on lines 152-153, "Hence, Filomena-like storms would be less probable in a future climate and would be less likely to produce such amounts of snowfall and strong winds" is without foundation and far too absolute.

**In the new version of the manuscript, with the new analysis provided, our conclusions about Filomena are: "Filomena-like storms in the factual period display higher slp yet cause more precipitation in central Spain, the region that suffered the highest impacts from the storm. Even though there are slightly more analogues in the coldest months, that is, January and February, there is a significant increase in the 2m temperature, making the snow at low altitudes less probable in a warmer climate. Given the reasonable quality of analogues, we can state that the results are in line with the expected climate change trends discussed in the previous section. However, since there is a shift in the distributions of ENSO conditioned to the analogues, we can not reject the hypothesis that ENSO variability has some influence on the analogues of Filomena" (LL214-219).**

Lines 166-167: The histograms in Figure 2(p) are so ragged that I find it entirely plausible that the changes shown might simply reflect sampling uncertainty. The sample size of 33 analogues does seem very small. How can you convince the reader that you have captured a real difference here?

**In view of the revised interpretation we have provided for Filomena and other events (see also our reply to the Reviewer's first general comment), we have replaced the sentence by the following:** "even though there are slightly more analogues in the coldest months, that is, January and February, there is a significant increase in the 2m temperature, making the snow at low altitudes less probable in a warmer climate." (LL 214-219)

Lines 274-276 say "the persistence index Θ (o) is higher in the recent period, indicating that recent cut-offs are more likely to stay stationary in Western Europe, leading to longer lasting precipitation events and potentially more intense floods." This is a remarkable and completely unjustified jump from an observed trend in one particular index to an unqualified attribution to climate change. So far as I am aware, there is no consensus whatsoever on how persistent summertime circulation regimes will respond to climate change, let alone the sort of regime that was conducive to the Westphalia floods.

**What we mean here is that when we have a cut-off such as the one observed for Westphalia floods, then it is more likely to stay stationary and therefore more likely to produce long lasting precipitation (hence floods). This statement is supported by the change in the Θ distribution that we show and therefore is a result of our analysis. We are not claiming that climate change will produce more stationary precipitation events in general. We have rephrased the sentence to avoid any misunderstandings. We also note that there are studies supporting specific responses of persistent summertime circulation regimes to climate change. For example, Kornhuber et al. (2021) state that: "models project an increase in weather persistence across the midlatitudes [...], with strongest signals over land-area", and argue that although model projections show a weak agreement one can use the models' performance against observational data to draw more robust conclusions.**

**After conditioning the research of analogs on the extended summer season only, we found no significant difference between the distribution of persistence among the analogs. This has been changed in the text: lines 327-329 "No significant changes are observed in the distributions of predictability D (Fig. 4n) and persistence Θ (Fig. 4o), nor in the predictability or persistence of the event itself relative to the circulation in the two periods".**

Lines 376-377 say "As we have discussed in Section 4.5.1, very intense hurricanes will become more frequent with climate change, and they will be more likely to undergo post-tropical transition." The language has considerably strengthened from that in Section 4.5.1, which was based on literature, and here, where it is unconditional. That increase in level of confidence is unjustified.

**We agree and we have toned down this sentence as: ""As we have discussed in Section 4.5.1, it is likely that very intense hurricanes will become more frequent with climate change, and they will probably be more likely to undergo post-tropical transition." (LL455-457)**

Lines 460-461 say "there is a general consensus that the jet stream will shift northward and therefore cut-off low will become slightly less probable on the Mediterranean sea". No reference is given for this statement, and it seems inconsistent with the AR6 Chapter 4 statement that "Substantial uncertainty and thus low confidence remain in projecting regional changes in Northern Hemisphere jet streams and storm tracks". Any such statement would need to be substantiated, especially for the autumn season in question.

**We have toned down this part and added a reference: " a recent study suggests that the jet stream will shift northward (Stendel et al. 2021) and therefore cut-off lows on the Mediterranean Sea may become slightly less frequent." LL 558-560.**

*Stendel, M., Francis, J., White, R., Williams, P. D., & Woollings, T. (2021). The jet stream and climate change. In Climate Change (pp. 327-357). Elsevier.*

Lines 511-514: This seems highly speculative. I found the result for this case study very surprising, and certainly worthy of further discussion if the implication is that certain dynamical situations can prevent the expected warming from anthropogenic climate change. (I have to say that I am finding it difficult to come up with a plausible physical mechanism.)

**This is a hypothesis that we make, partly based on the rich literature (and heated discussions) concerning the Warm Arctic - Cold Eurasia pattern and its dynamical drivers. The part that is indeed highly speculative is that the Cold Eurasia part of the pattern may then reflect on the Scandinavian cold spells occurring under easterly advection - something which we find plausible but that we are not aware has been systematically studied in the literature. In the original text, we only grazed this aspect by referring to the well-known study by Cohen and colleagues, but we agree that it would be interesting to extend our reasoning further. In the revised text, we have both phrased more clearly that this is a hypothesis rather than a robust result of our analysis and provide a broader context for this statement. We have anchored the extended discussion in the Warm Arctic - Cold Eurasia literature, including studies that have linked this pattern to the occurrence of extreme cold spells (e.g. Ye and Messori, 2020). See LL 237-252 for the changes in the text.**

**Ye, K., & Messori, G. (2020). Two Leading Modes of Wintertime Atmospheric Circulation Drive the Recent Warm Arctic–Cold Eurasia Temperature Pattern, *J. Clim.*, 33(13), 5565-5587**

Typos: you sometimes say ECT when you mean ETC

**Thank you for spotting this. We have proof-readed with greater care our revised text.**

**Answers Reviewer 2**

The paper uses circulation analogues to determine to what extent recently observed extreme events may be influenced by the warming between the first half of the ERA data since 1950 and the more recent period. The analysis is well conducted and the results are explained well. I clicked major revision because i feel quite strongly about some of my comments . Overall, i really enjoyed reading a careful analogue analysis of several interesting events and once the concerns of the reviewers are addressed this will be a very nice addition to the literature. (note i made my comments first and then checked what the other reviewer commented)

**We thank the Reviewer for their positive outlook on our submission, and recognise that some of the statements in our original text require rephrasing or qualifying. We provide detailed replies (in bold) to the Reviewer's comments below. Additionally, we note that ECMWF is an updated version of the ERA5 back-extension which we have downloaded. All the analysis has been updated to use these new data.**

Major issues that need to be addressed

1) The way IPCC conclusions are referred to in the introduction is very misleading. I read some of the referred to conclusions with surprise. I think the only way to sensible quote ipcc conclusions is to quote them verbally and in ". Everything else can be misleading and does an injustice to the assessment process and approval plenary process. So this section needs to be reworked, clarifying what conclusions are IPCC conclusions and what the authors infer based on their reading. i note that the previous reviewer requested this as well

**We accept the criticism that we have been not precise in quoting the IPCC AR6 report here. To avoid any misstatements of the IPCC conclusions, in the revised version of the study, we have followed the reviewer's suggestion and preferentially use verbatim quotes to support our references to the IPCC AR6 report.**

2) There is some statistical lack of rigour in some of the statements. Firstof, as you say, the human influence is not 0 in the first part of your analysis period, and secondly, decadal variability can be important on the timeline you investigate. You need to acknowledge this more clearly in conclusions and you also need to discuss this as a caveat more clearly. nothing in your analysis allows you to determine if a change is anthropogenic, natural, or decadal variability. Some changes you see are consistent with well understood trend attribution results (which you could refer to, and that is what the US National academies report 2016 on event attribution recommended!) you should do this a bit more. examples are warming and moistening on large scales. But in some cases there is no way to know and you should acknowledge that. Also some statements are phrased rather absolute for example the change in seasonality in events which based on the diagrams shown look quite tenuous.

**We do agree that some of the statements provided in the paper needed for a more solid statistical support. In the new version of the manuscript, significant changes**

between distributions shown in panels m,n,o are assessed using a wilksum test. The figures now report the p-values associated with the test and the discussion critically reviews these results.

While we agree that we do not separate the different possible sources of changes in atmospheric patterns in our analysis, we would like to highlight that, already in 2011, a leading climate scientist such as Kevin Trenberth argued that: "Past attribution studies of climate change have assumed a null hypothesis of no role of human activities... I argue that because global warming is "unequivocal" and 'very likely' caused by human activities, the reverse should now be the case. The task, then, could be to prove there is no anthropogenic component to a particular observed change in climate" (Trenberth, 2011). While we understand the Reviewer's concerns, we thus believe that there are also arguments for making statements about anthropogenic influences on observed circulation changes in the absence of a full statistical analysis of the potential role of natural variability. We have additionally conducted an analysis on the possible roles of ENSO and AMO in modulating the analogues we analyze. Specifically, we analyze the distributions of the ENSO and AMO indices corresponding to analogues of each event in the factual and counterfactual periods. If the null hypothesis that the two distributions do not differ between the two periods is rejected, it is not possible to exclude that thermodynamic or dynamic differences in the analogues are due to the corresponding mode of natural variability, rather than anthropogenic forcing. On the other hand, if it is not possible to reject the null hypothesis of equal distribution, observed changes in analogues can be attributed to human activity.

We have additionally modified several passages of our text to acknowledge the possible role of non-anthropogenic sources of variability, for example on LL146-155, LL 650-664 and within the discussion of each extreme event.

Trenberth, K. E. (2011). Attribution of climate variations and trends to human influences and natural variability. *Wiley Interdisciplinary Reviews: Climate Change*, *2*(6), 925-930.

3) I am a bit nervous about using analogues across all seasons. the feedback to hot extremes, for example, will be different in the hot season, as will be the influence of SSTs. Also discussion of circulation variability between the periods is highly uncertain. this needs to be reworked a bit, and unless you have tested (considering decadal variablity!) if a change is significant you need to be careful. i think at most you can conclude that there is a significant difference between your analysis period, but you cant attribute it to forcing or variability. But i am not sure you can even conclude that in some cases where you state it.

We believe the reviewer is raising two important points in this comment. The first concerns the seasonality of the analogues: we have now repeated the analysis for each of the extreme events conditional on season. We have picked four months (DJFM, MAMJ, JJAS, SOND) overlapping to allow for the fact that there is not a sharp cut-off between the different seasonal circulations and the different physical processes modulating extremes in different periods of the year.  For several of the

extremes, the seasonal shifts highlighted by the full year analysis are confirmed by the seasonal analysis, for example medicane Apollo showed a shift to more frequent analogues in September in the factual period and the analysis during SOND also shows an increase in analogues in September (see figures R1,R2 below). We note however that other aspects of our results such as the connection between analogues and low frequency modes of natural variability show a seasonal dependence. We agree with the issue outlined by the reviewer that the full-year analysis may conflate different physical processes contributing to the extremes and result in misleading interpretations. We have thus decided to only show the seasonal analyses in the revised manuscript.

The second part of the comment concerns the possible role of natural variability in explaining the differences we find between factual and counterfactual periods. As stated in our reply to comment 2), we have implemented an additional analysis and several textual changes in the manuscript to address this.

[Figure]

Figure R1 : Full year analysis for Medicane Apollo

[Figure]

Figure R2 : SOND analysis for Medicane Apollo.

4) i am not sure about the discussion of probabilistic event attribution vs your approach. the observations only approach is very useful and complementary to the big model based pdfs but you shouldn t claim that these do not take into account the processes leading to the events. It almost sounds like you read no paper from the modelling approach side - hey usually analyze synoptic situation and discuss contributions. I feel quite strongly this needs to change as the discussion of event attribution approaches has been surprisingly emotional in the literature, advocating the chosen approach by misrepresenting what people using other approaches do. This is unscientific and unhelpful. I find the world weather attribution approach nice as it uses multiple approaches and gains strength from doing so.

**We did not mean to discredit other attribution approaches, and indeed in our conclusions we clearly state that: "We [..] see our analysis as complementing statistical approaches by providing insights on the possible changes over time of the dynamics underlying extreme events from a dynamical perspective". We further refer**

twice to results from the World Weather Attribution initiative to contextualize the events we are studying.  In the new version of the manuscript we make very clear that in LL 664-677.  Furthermore, we would like to point the reviewer also this nice perspective article in Science, which shows how our technique complements a wealth of new studies which all aim at complementing the traditional statistical attribution approaches:

[https://www.science.org/content/article/record-shattering-events-spur-advances-in-tying-climate-change-to-extreme-weather](https://www.science.org/content/article/record-shattering-events-spur-advances-in-tying-climate-change-to-extreme-weather)

**and not to substitute them.**

Minor comments

Title: this really deals with European extremes and one American extreme. At most, this is about extratropical NH extremes. Given the publication bias towards developed world extremes you should not use a title that implicitly claims this to be a global analysis - a lot of extremes have occurred elsewhere and you are not talking about those.

**While we definitely agree with the reviewer on our selection bias towards extremes affecting the countries we live or we are originally from (coinciding with those of the developed world), we would like to keep the title as it is, since it already contains the wording "some" which implies that our study is not comprehensive of all events**

p. 1 l 10 i think its analogues? not analogs - this may be elsewhere

**"Analog" and "analogue" are both valid alternative spellings. WCD allows for either American or British spelling upon typesetting, so we chose not to change this.**

p. 2: the introduction absolutely needs to change please - either quote papers for your summary of what they say, or the IPCC chapters verbatim. dont put words into the mouth of the ipcc.

**As mentioned in our reply to major comment #1, to avoid any misstatements of the IPCC's conclusions, in the revised version of the study we will follow the reviewer's suggestion and preferentially use verbatim quotes to support our references to the IPCC AR6 report.**